# Variational Learning of Fractional Posteriors

**Kian Ming A. Chai** [1]   **Edwin V. Bonilla** [2]

## Abstract

We introduce a novel one-parameter variational objective that lower bounds the data evidence and enables the estimation of approximate fractional posteriors. We extend this framework to hierarchical construction and Bayes posteriors, offering a versatile tool for probabilistic modelling. We demonstrate two cases where gradients can be obtained analytically and a simulation study on mixture models showing that our fractional posteriors can be used to achieve better calibration compared to posteriors from the conventional variational bound. When applied to variational autoencoders (VAEs), our approach attains higher evidence bounds and enables learning of high-performing approximate Bayes posteriors jointly with fractional posteriors. We show that VAEs trained with fractional posteriors produce decoders that are better aligned for generation from the prior.

## 1. Introduction

Exact Bayesian inference is intractable for most models of interest in machine learning. Variational methods (Jordan et al., 1999; Minka, 2001; Opper & Winther, 2005; Blei et al., 2017) address this by casting the required integration as optimisation. These methods have two objectives: to estimate the marginal likelihood or data evidence (MacKay, 2003) for model comparison or model optimisation; and to obtain an approximate Bayes posterior for prediction.

The widely used evidence lower bound (ELBO) (Jordan et al., 1999) often leads to underestimated uncertainty and suboptimal posterior calibration (Wang & Titterington, 2005; Bishop, 2006; Yao et al., 2018). These deficiencies can be compounded by challenges inherent to general Bayesian modelling, such as misspecification. Fractional

posteriors (Grünwald & van Ommen, 2017; Bhattacharya et al., 2019) have emerged as a generalization of Bayesian inference to address these. Unlike the Bayes posterior, which fully incorporates the likelihood, a fractional posterior has an exponent that weighs the likelihood to temper its influence. This approach has been shown to enhance robustness in misspecified models and has strong connections to PAC-Bayesian bounds (Bhattacharya et al., 2019), which control generalization error in statistical learning.

This work introduces a new variational framework that generalises conventional variational inference (VI) by allowing the approximation of fractional posteriors, enabling improved posterior flexibility and calibration. As in standard VI based on ELBO maximisation, our approach provides a lower bound on the marginal likelihood and extends to hierarchical construction (Ranganath et al., 2016) and Bayes posteriors. Thus it offers a flexible trade-off between evidence maximization and posterior calibration, bridging the gap between standard VI and fractional Bayesian inference.

We explore both analytical and empirical insights into variational learning of fractional posteriors. First, we identify cases where gradients can be derived analytically, eliminating the need for gradient estimators (Roeder et al., 2017). Next, we perform a simulation study on mixture models, demonstrating that fractional posteriors achieve better-calibrated uncertainties compared to conventional VI. Finally, we consider variational autoencoders (VAEs) (Kingma & Welling, 2014), showing that fractional posteriors not only give higher evidence bounds but also enhance generative performance by aligning decoders with the prior—a known issue in standard VAEs (Dai & Wipf, 2019).

This work advances the field of approximate Bayesian inference with a theoretically grounded and empirically validated approach to fractional variational inference. It demonstrates that fractional posteriors can improve model calibration and yield better generative models, and it offers an alternative to standard variational inference and learning approaches.

**Notation**   We use letter $p$ for model distributions and letters $q$ and $r$ for approximate distributions. The tilde (˜) accent is for the unnormalised version of the distribution that it modifies; and the asterisk (*) superscript is for the optimised versions. The unaccented letter $Z$ is for the nor-

[1]DSO National Laboratories, Singapore [2]CSIRO's Data61, Australia. Correspondence to: Kian Ming A. Chai <ckianmin@dso.org.sg>, Edwin V. Bonilla <edwin.bonilla@data61.csiro.au>.

*Proceedings of the $42^{nd}$ International Conference on Machine Learning*, Vancouver, Canada. PMLR 267, 2025. Copyright 2025 by the author(s).

malising constant, and $\tilde{Z}$ is for an arbitrary scaling constant. We reserve *ELBO* for the conventional bound (Jordan et al., 1999). Boldfaces are used only when needed to distinguish vectors from scalars. We omit the *approximate* qualifier from *approximate posterior*, unless the context requires it. When comparing evidence bounds, we use the adjective *tighter* if we can ascertain that the bound is closer to a fixed evidence value; and we use the adjective *higher* if bounds cannot be compared for tightness because their corresponding fixed evidences are different. The latter is limited to sections 5.2 and 5.3 when we optimise the likelihood of the model. Table 3 in section A lists the bounds in this paper.

## 2. A Lower Bound with Hölder's Inequality

The log-evidence $\mathcal{L}_{\mathrm{evd}} \stackrel{\text{def}}{=} \log p(D)$ of data $D$ for a generative model involving an auxillary variable $z$ is bounded by the variational Rényi (lower) bound $\mathcal{L}_\alpha^{\mathrm{R}}$ (Li & Turner, 2016, Theorem 1) for $\alpha > 0$:

$$\mathcal{L}_{\mathrm{evd}} \geq \mathcal{L}_\alpha^{\mathrm{R}} \stackrel{\text{def}}{=} \frac{1}{1-\alpha} \log \int q(z)\,(p(D,z)/q(z))^{1-\alpha}\,\mathrm{d}z.$$

The Kullback-Leibler divergence (KL, $\alpha \to 1$) is the only case where the chain rule of conditional probability holds exactly to get the conventional ELBO $\mathcal{L}_{\mathrm{ELBO}} \stackrel{\text{def}}{=} \int q(z) \log p(D|z)\mathrm{d}z - \int q(z) \log(q(z)/p(z))\mathrm{d}z$. This involves the expected log conditional likelihood (first term) that encourages data fitting and the KL term between the approximate posterior $q(z)$ and the prior $p(z)$ (second term) that acts as a regulariser to bias the posterior towards the prior. This decomposition is not possible for other values of $\alpha$, so $\mathcal{L}_\alpha^{\mathrm{R}}$ generally cannot be expressed in such terms.

To this end, we revert to the original log-evidence $\mathcal{L}_{\mathrm{evd}}$ and apply Hölder's inequality (Rogers, 1888) in the manner of $\left(\mathbb{E}\left[|X|^{1/\beta}\right]\right)^\beta \geq \mathbb{E}[|XY|] / \left(\mathbb{E}\left[|Y|^{1/\gamma}\right]\right)^\gamma$, where $\beta + \gamma = 1$ and $\beta, \gamma \in (0,1)$, with

$$\mathbb{E}[\cdot] \stackrel{\text{def}}{=} \int p(z) \cdot \mathrm{d}z, \quad X \stackrel{\text{def}}{=} p(D|z)^\beta, \quad Y \stackrel{\text{def}}{=} \tilde{q}(z)/p(z)$$

to obtain

$$\begin{aligned}
\mathcal{L}_{\mathrm{evd}} &= \frac{1}{\beta} \log \left(\int p(z)\,p(D|z)\mathrm{d}z\right)^\beta \\
&\geq \frac{1}{\beta} \log \frac{\int \tilde{q}(z)\,p(D|z)^\beta\,\mathrm{d}z}{\left(\int p(z)\,(\tilde{q}(z)/p(z))^{1/\gamma}\,\mathrm{d}z\right)^\gamma} \\
&= \frac{1}{1-\gamma} \log Z_{\mathrm{d}} - \frac{\gamma}{1-\gamma} \log Z_{\mathrm{c}} \quad \stackrel{\text{def}}{=} \mathcal{L}_\gamma, \quad (1)
\end{aligned}$$

where we have expressed $\beta$ in $\gamma$, and we have data-fitting and regularisation (complexity) terms

$$\tilde{q}_{\mathrm{d}}(z) \stackrel{\text{def}}{=} \tilde{q}(z)\,p(D|z)^{1-\gamma} \quad \tilde{q}_{\mathrm{c}}(z) \stackrel{\text{def}}{=} \tilde{q}(z)^{1/\gamma}\,p(z)^{1-1/\gamma}$$

$$Z_{\mathrm{d}} \stackrel{\text{def}}{=} \int \tilde{q}_{\mathrm{d}}(z)\mathrm{d}z \qquad Z_{\mathrm{c}} \stackrel{\text{def}}{=} \int \tilde{q}_{\mathrm{c}}(z)\mathrm{d}z.$$

The derivation does not require $\tilde{q}(z)$ to be a distribution. However $Y$ must be in $L^\gamma$ to apply the Hölder's inequality, that is, $\operatorname{supp} \tilde{q} \subseteq \operatorname{supp} p$. Hence, the $\tilde{q}$ must be consistent with the prior $p$, a desideratum. If $\tilde{q}(z)$ is a distribution $q(z)$, that is, $\int q(z)\mathrm{d}z = 1$, then the second term of the bound is the Rényi divergence $\mathrm{D}_{1/\gamma}[q(z)\|p(z)]$.

Our objective $\mathcal{L}_\gamma$ in eq. (1) can be seen as an example of the *generalized variational inference* framework (Knoblauch et al., 2022). However, it is uniquely derived as a lower bound to the log-evidence, so it naturally encodes the Occam's razor principle and can be used for model optimisation (MacKay, 2003, Chapter 28). We can relate this objective to the conventional ELBO (Lemma A.3 shows that $\lim_{\gamma \to 1} \mathcal{L}_\gamma = \mathcal{L}_{\mathrm{ELBO}}$ using the L'Hôpital's rule and the convergence of Rényi to KL divergence) and also provide two related upper bounds (Lemmas A.1 and A.2).

### 2.1. Fractional Posteriors

The optimal $\mathcal{L}_\gamma$ is tight at $\tilde{q}^*(z) = p(D|z)^\gamma p(z)/\tilde{Z}$, where $\tilde{Z}$ can be the normalising constant:

$$\begin{aligned}
\mathcal{L}_\gamma^* &= \frac{1}{1-\gamma} \log \tilde{Z}^{-1} \int p(D|z)\,p(z)\mathrm{d}z \\
&\quad - \frac{\gamma}{1-\gamma} \log \tilde{Z}^{-1/\gamma} \int p(D|z)p(z)\mathrm{d}z \\
&= \log \int p(D|z)p(z)\mathrm{d}z \quad \equiv \mathcal{L}_{\mathrm{evd}}.
\end{aligned}$$

Since $\mathcal{L}_\gamma$ is a lower bound, this already proves the optimality of $\tilde{q}^*$ (section A.1 gives a variational derivation). In contrast, the gap between $\mathcal{L}_{\mathrm{evd}}$ and $\mathcal{L}_\alpha^{\mathrm{R}}$ is the Rényi divergence $\mathrm{D}_\alpha[q(z)\|p(z|D)]$ (section A.2), so $\mathcal{L}_\alpha^{\mathrm{R}}$'s optimal $q(z)$ is the exact Bayes posterior $p(z|D)$. Nonetheless, $\mathcal{L}_\gamma$ is related to $\mathcal{L}_\alpha^{\mathrm{R}}$ by suitable change of distributions (section A.3).

The above shows that the bound is tight at a *fractional posterior* where the data-likelihood is weighted with $\gamma \in (0,1)$ (Bhattacharya et al., 2019). This is more generally known as the Gibbs posterior (Zhang, 1999; Alquier et al., 2016), the power posterior (Friel & Pettitt, 2008), or the tempered posterior (Pitas & Arbel, 2024). As mentioned in the introduction, fractional posteriors are related to robustness in misspecified models (Bhattacharya et al., 2019).

We have obtained the fractional posterior directly from optimising $\mathcal{L}_\gamma$, which allows approximations to the unnormalised fractional posterior via optimisation within an assumed family of non-negative functions. It is achieved without relying on PAC-Bayes or modifying the likelihood, and it follows from optimising a lower bound on $\mathcal{L}_{\mathrm{evd}}$. It is an alternative to the approach by Alquier et al. (2016).

### 2.2. On the choice of $\gamma$

It is generally futile to seek a $\gamma^*$ giving the tightest bound: for a given probabilistic model and a fixed posterior $q$, $\gamma^*$

depends on $q$. For, if $q$ is close to the exact Bayes posterior, then $\gamma^* = 1$; if $q$ is close to an exact fractional posterior, then $\gamma^*$ is that fraction. The situation is the same if we learn $q$ within a family $\mathcal{Q}$. If $\mathcal{Q}$ contains all the exact posteriors, Bayes and fractionals, then all values of $\gamma$s are optimal because all give $\mathcal{L}_{\text{evd}}$ after optimisation. If, however, $\mathcal{Q}$ can approximate only certain fractional posteriors well, then the corresponding $\gamma$s will give the tightest bounds.

Nonetheless, if one applies approximate inference analytically to a problem that is specified with an explicit prior, such as the normal distribution, it is typical to choose $\mathcal{Q}$ to be in the same family as the prior, such that $\mathcal{Q}$ includes the prior and its neighbourhood. In this case, we expect optimising with small $\gamma$ to give consistently tighter bounds. This is shown in section 5.1 empirically.

In a similar fashion, for challenging data sets for which we use neural networks, we expect smaller $\gamma$ to give tighter bounds for simpler neural networks that can better approximate the prior and the fractional posteriors than the Bayes posterior. This is supported in section C.4 empirically.

Considerations other than bounds may influence the choice of $\gamma$. In section 5.1, we use calibration; in section 5.3, we want a posterior that is close to the prior.

### 2.3. Extensions to Hierarchical Constructions

We give two extensions to $\mathcal{L}_\gamma$ to allow for more expressive fractional and Bayes posteriors using mixing. We show in sections A.5.1 and A.5.2 that degeneracy of the mixing distribution is not necessary for optimality, in contrast to the case for ELBO (Yin & Zhou, 2018, Proposition 1).

#### 2.3.1. FRACTIONAL POSTERIOR

Let $\tilde{q}(z) \stackrel{\text{def}}{=} \int \tilde{q}(z|u)q(u)\mathrm{d}u$ be a hierarchical model of the posterior distribution using mixing variable $u$. Jensen's inequality for convexity of powers above unity gives

$$
\begin{aligned}
Z_{\text{c}} &= \int \left( \int \tilde{q}(z|u)q(u)\mathrm{d}u \right)^{1/\gamma} p(z)^{1-1/\gamma}\mathrm{d}z \\
&\leq \int \left( \int \tilde{q}(z|u)^{1/\gamma}q(u)\mathrm{d}u \right) p(z)^{1-1/\gamma}\mathrm{d}z \\
&= \int \left( \int \tilde{q}(z|u)^{1/\gamma-1}\tilde{q}(z|u)\, q(u)\mathrm{d}u \right) p(z)^{1-1/\gamma}\mathrm{d}z \\
&= \iint \left( \tilde{q}(z|u)/p(z) \right)^{1/\gamma-1} \tilde{q}(z|u)\mathrm{d}z\, q(u)\mathrm{d}u \quad (2)
\end{aligned}
$$

We may substitute this into eq. (1) to obtain another bound, which we shall call $\mathcal{L}_\gamma^{\text{h}}$. This lower bound allows Monte Carlo estimates of the integral by only using samples from the posterior $\tilde{q}(z|u)$ (see section 4).

A similar approach has been used to lower bound the ELBO (Yin & Zhou, 2018, Theorem 1). There, the optimal for $q(u)$ is known to be the delta distribution located for optimal $q(z|u)$ (Yin & Zhou, 2018, Proposition 1). However, deviations from this property may happen in practice (see sections B.2 and C.4).

#### 2.3.2. BAYES POSTERIOR

We can bound the data term in $\mathcal{L}_\gamma$ using Jensen's inequality with another variational distribution $r(z)$:

$$
\begin{aligned}
\log Z_{\text{d}} \geq (1-\gamma) \int r(z)\log p(D|z)\mathrm{d}z \\
- \int r(z)\log \frac{r(z)}{\tilde{q}(z)}\mathrm{d}z
\end{aligned}
$$

so that logarithm over the product of likelihoods becomes a sum over the logarithms of each likelihood, and we may sample over the data points. Section 3.1.3 gives a different but more specific bound for the mixture model.

The above bound is exact at $r^*(z) \propto \tilde{q}(z)\, p(D|z)^{1-\gamma}$. If $\tilde{q}(z)$ is also optimal, then $r^*(z) \propto p(z)p(D|z)$, which is the Bayes posterior. Combining the above bound with eq. (1) gives a bound on $\mathcal{L}_{\text{evd}}$ involving both KL and Rényi divergences. We shall denote this bound by $\mathcal{L}_\gamma^{\text{b}}$.

If we fix $r(z)$ regardless of its optimality, and then optimise for $\tilde{q}(z)$, we obtain $\tilde{q}^*(z) \propto r(z)^\gamma p(z)^{1-\gamma}$ interpolating between the fixed $r(z)$ and the model prior $p(z)$. This shows that $r(z)$ has a constraining effect on $\tilde{q}(z)$, so the fractional posteriors approximated by $\mathcal{L}_\gamma^{\text{b}}$ are in general different from those approximated by using $\mathcal{L}_\gamma$. In particular, *if $r(z)$ itself is a fractional posterior with fraction $\gamma'$*, then $\tilde{q}(z)$ has at best fraction $\gamma'\gamma$. This is shown empirically in section 5.2.1.

For a fixed $r(z)$, $\mathcal{L}_\gamma^{\text{b}}$ is upper bounded by $\mathcal{L}_{\text{ELBO}}$ (see Lemma A.4), so $\mathcal{L}_\gamma^{\text{b}}$ in itself has limited use. However, we can use it with the hierarchical posterior model (section 2.3.1) to give more expressive posteriors. For this, we have to go beyond just applying the hierarchical model on $r(z)$ because this will result in a degenerate mixing distribution for $r(z)$, since the terms involved are exactly the same as in ELBO. To prevent degeneracy, we apply hierarchical model to *both* $\tilde{q}(z)$ and $r(z)$, with the *same* mixing distribution $q(u)$. That is, $\tilde{q}(z) \stackrel{\text{def}}{=} \int \tilde{q}(z|u)q(u)\mathrm{d}u$ and $r(z) \stackrel{\text{def}}{=} \int r(z|u)q(u)\mathrm{d}u$. Under this setting, we apply eq. (2) on the Rényi divergence term and the convexity on the KL term to obtain a bound we call $\mathcal{L}_\gamma^{\text{bh}}$ (see section A.4):

$$
\begin{aligned}
\mathcal{L}_{\text{evd}} \geq \iint r(z|u)q(u)\log p(D|z)\, \mathrm{d}z\mathrm{d}u \\
- \frac{1}{1-\gamma}\iint r(z|u)q(u)\log \frac{r(z|u)}{\tilde{q}(z|u)}\mathrm{d}z\mathrm{d}u \\
- \frac{\gamma}{1-\gamma}\log \iint \tilde{q}(z|u)q(u)\left( \frac{\tilde{q}(z|u)}{p(z)} \right)^{1/\gamma-1} \mathrm{d}z\mathrm{d}u.
\end{aligned}
$$

# 3. Learning

Let $\tilde{q}$ be parameterised by $\theta$. Under regularity conditions,

$$\frac{\partial \mathcal{L}_\gamma}{\partial \theta} = \frac{1}{1-\gamma} \int (q_{\mathrm{d}}(z) - q_{\mathrm{c}}(z)) \frac{\partial \log \tilde{q}(z)}{\partial \theta} \mathrm{d}z,$$

where $q_{\mathrm{d}}(z) \stackrel{\text{def}}{=} \tilde{q}_{\mathrm{d}}(z)/Z_{\mathrm{d}}$ and $q_{\mathrm{c}}(z) \stackrel{\text{def}}{=} \tilde{q}_{\mathrm{c}}(z)/Z_{\mathrm{c}}$ are normalised distributions, and we have used the log-derivative trick $\partial \log \tilde{q}/\partial \theta = (1/\tilde{q})(\partial \tilde{q}/\partial \theta)$. At the optimal $\tilde{q}^*$, both $q_{\mathrm{c}}(z)$ and $q_{\mathrm{d}}(z)$ equates the exact Bayes posterior $p(z|D)$. Setting the gradient to zero entails matching the expectations of the gradients of $\log \tilde{q}$ under $q_{\mathrm{c}}$ and $q_{\mathrm{d}}$.

Gradients for $\mathcal{L}_\gamma^{\mathrm{b}}$ and $\mathcal{L}_\gamma^{\mathrm{bh}}$ can be similarly expressed.

## 3.1. Case Studies

We study three cases of applying $\mathcal{L}_\gamma$ analytically. The first case where exact inference is possible is illustrative. The other cases, where exact inference is not, demonstrate where using $\mathcal{L}_\gamma$ can be useful.

### 3.1.1. EXPONENTIAL FAMILY

Consider $D$ to be a collection of $n$ independent data $\{x_1, \ldots, x_n\}$ in the exponential family with the conjugate prior setting: $p(x_i|z) = h(x_i) \exp \left( z^{\mathrm{T}} t(x_i) - a(z) \right)$; $p(z|\nu, \kappa) = g(\nu, \kappa) \exp \left( z^{\mathrm{T}} \nu - \kappa a(z) \right)$; and $\tilde{q}(z|\mu, \lambda) = \exp \left( z^{\mathrm{T}} \mu - \lambda a(z) \right)$, with $t$ being the sufficient statistic, $z$ the natural parameter, $a$ the log-partition function; $\nu$ and $\kappa$ the parameters for the prior; and $\mu$ and $\lambda$ the parameters for the posterior. Then $\partial \log \tilde{q}(z)/\partial \mu = z$ and $\partial \log \tilde{q}(z)/\partial \nu = -a(z)$, and

$$q_{\mathrm{c}}(z) \propto \exp \left( z^{\mathrm{T}} \left( \mu/\gamma + (1 - 1/\gamma)\nu \right) - k(\kappa)a(z)/\gamma \right)$$
$$q_{\mathrm{d}}(z) \propto \exp \left( z^{\mathrm{T}} \left( \mu + (1 - \gamma) \sum_{i=1}^n t(x_i) \right) - k(n)a(z) \right),$$

where $k(\bullet) \stackrel{\text{def}}{=} \lambda + (1 - \gamma)\bullet$. The sufficient statistics for conjugate distributions are $z$ and $-a(z)$, so

$$\mathbb{E}_{q_{\mathrm{c}}}[z] = \mu/\gamma + (1 - 1/\gamma)\nu \qquad \mathbb{E}_{q_{\mathrm{c}}}[-a(z)] = k(\kappa)/\gamma$$
$$\mathbb{E}_{q_{\mathrm{d}}}[z] = \mu + (1 - \gamma) \sum_i^n t(x_i) \qquad \mathbb{E}_{q_{\mathrm{d}}}[-a(z)] = k(n).$$

Zeroing gradients $\partial \mathcal{L}_\gamma/\partial \mu$ and $\partial \mathcal{L}_\gamma/\partial \lambda$ gives the following parameters for $\tilde{q}$ as expected:

$$\mu = \nu + \gamma \sum_{i=1}^n t(x_i) \qquad \lambda = \kappa + \gamma n.$$

The parameters interpolate between the prior and the Bayes posterior, as a consequence that exact inference is achievable. This is not true for more general models and approximate inference may be required.

### 3.1.2. MULTINOMIAL DATA WITH GAUSSIAN PRIOR

In the previous case where exact posteriors can be obtained, it is not necessary to derive the gradients, since we already know the functional form of these posteriors.

Where the exact fractional posterior could be complicated, we assume a functional form for the approximate posterior. Consider the model with a multinomial logit likelihood for $C$ classes and a standard Gaussian prior:

$$p(x|z) = \frac{\exp z_x}{\sum_{c=1}^C \exp z_c}; \quad p(z) = \prod_{c=1}^C \frac{1}{\sqrt{2\pi}} \exp -\frac{z_c^2}{2}.$$

For $n$ data points, we choose $1/\gamma = 1 + 1/n$ and let

$$\tilde{q}(z) = \left( \prod_{c=1}^C \mathcal{N}(z_c|\mu_c, \sigma_c^2) \right) \left( \sum_{c=1}^C \exp z_c \right)^{n/(n+1)}.$$

Then

$$q_{\mathrm{c}}(z) \propto \left( \prod_{c=1}^C \mathcal{N}(z_c|m_c, s_c^2) \right) \left( \sum_{c=1}^C \exp z_c \right)$$
$$= \sum_{c=1}^C \exp z_c \left( \prod_{c'=1}^C \mathcal{N}(z_{c'}|m_{c'}, s_{c'}^2) \right)$$
$$q_{\mathrm{d}}(z) \propto \left( \prod_{c=1}^C \mathcal{N}(z_c|\mu_c, \sigma_c^2) \right) \prod_{i=1}^n \exp \left( z_{x_i}/(n+1) \right)$$
$$\propto \prod_{c=1}^C \mathcal{N} \left( z_c|\mu_c + \frac{n_c \sigma_c^2}{n+1}, \sigma_c^2 \right),$$

where

$$m_c \stackrel{\text{def}}{=} \frac{\mu_c(n+1)}{n+1-\sigma_c^2} \qquad s_c^2 \stackrel{\text{def}}{=} \frac{n\sigma_c^2}{n+1-\sigma_c^2},$$

and $n_c \stackrel{\text{def}}{=} \sum_{i=1}^n \delta(c, x_i)$ is the number of data points of class $c$. The last expression of $q_{\mathrm{c}}$ has normalising constant $\sum_{c=1}^C \exp(m_c + s_c^2/2)$, and the last expression for $q_{\mathrm{d}}$ is normalised because it is a product of independent Gaussian distributions. The gradients with respect to the parameters and the required expectations are given in section A.6. This is an example where we need only the unnormalised density $\tilde{q}$ during optimisation.

### 3.1.3. MIXTURE MODEL

A common model in the Bayesian literature is the mixture model. For $n$ samples $\{x_i\}_{i=1}^n$ and $K$ components with parameters $\{u_k\}_{k=1}^K$ independently drawn from $p(u_k)$, the evidence is $\sum_c p(c) \int p(u) \prod_{i=1}^n p(x_i|c_i, u) \mathrm{d}u$ (Blei et al., 2017, Equation 9), where $c_i \in \{1, \ldots, K\}$ is the latent assigned cluster for the $i$th sample, and the $c_i$s are independent. The outer sum over $K^n$ cluster assignments makes exact inference intractable.

Assume a mean-field approximation for the posterior: $q(u, c) = \prod_{k=1}^K q(u_k) \prod_{i=1}^n q(c_i)$. Like ELBO, we may apply the $\mathcal{L}_\gamma^{\mathrm{b}}$ bound to convert the innermost product to an outer sum for the likelihood term. Alternatively, we can first apply $\mathcal{L}_\gamma$ for the variational posterior $q(u)$ of the component means, and then the ELBO for the cluster assignments $c$, so that the $Z_{\mathrm{d}}$ term in eq. (1) is lower bounded by

$$\int q(u) \prod_{i=1}^n \prod_{c_i=1}^K (p(x_i|c_i, u)p(c_i)/q(c_i))^{(1-\gamma)q(c_i)} \mathrm{d}u.$$

Define the variational parameters $\varphi_{ik} \overset{\text{def}}{=} q(c_i = k)$, $i = 1, \ldots, n$, $k = 1, \ldots, K$. Simplifying with the mean-field independence assumption, the lower bound is (section A.7)

$$\sum_{k=1}^{K} \left( \frac{1}{1-\gamma} \sum_{i=1}^{n} \log \int q(u_k) p(x_i|u_k)^{(1-\gamma)\varphi_{ik}} \mathrm{d}u_k \right.$$
$$- \sum_{i=1}^{n} \varphi_{ik} \log \left( \varphi_{ik}/p(c_i = k) \right)$$
$$\left. - \frac{\gamma}{1-\gamma} \log \int q(u_k)^{1/\gamma} p(u_k)^{1-1/\gamma} \mathrm{d}u_k \right).$$

Identifying terms with $\mathcal{L}_\gamma$, optimality gives $q(u_k) \propto p(u_k) \prod_{i=1}^{n} p(x_i|u_k)^{\varphi_{ik}\gamma}$. For $\varphi_{ik}$, we first define distributions $q_i(u_k) \propto q(u_k) p(x_i|u_k)^{(1-\gamma)\varphi_{ik}}$, which introduces the proportion of the $i$th sample omitted in $q(u_k)$. Then $\varphi_{ik} \propto p(c_i) \exp\left( \mathbb{E}_{q_i}[\log p(x_i|u_k)] \right)$ — in this way, $q(c_i)$ approximates the Bayes posterior by using the full contribution of likelihood due to $x_i$. A full ELBO solution is obtained by setting $\gamma = 1$ for the updates; and in particular, $q_i(u_k) \equiv q(u_k)$ for all $i$.

If each conditional likelihood $p(x_i|u_k)$ is in the exponential family and the prior $p(u_k)$ is conjugate to it, then $q(u_k)$ is in the same conjugate family that have the sufficient statistics of the data weighted by $\varphi_{ik}$ and $\gamma$. Consider the Gaussian mixture model of Blei et al. (2017, §2.1), where the component priors are identically normal with mean zero and variance $\sigma^2$; the assignment priors are identically uniform; and the likelihood is unit variance normal centered at $u_k$. Then $q(u_k)$ is normal with mean and variance

$$\frac{\gamma \sum_{i=1}^{n} \varphi_{ik} x_i}{1/\sigma^2 + \gamma \sum_{i=1}^{n} \varphi_{ik}} \quad \text{and} \quad \frac{1}{1/\sigma^2 + \gamma \sum_{i=1}^{n} \varphi_{ik}}.$$

The same expressions are obtained from the approximate Bayes posterior $r(u_k)$ and the prior $p(u_k)$ using $r(u_k)^\gamma p(u_k)^{1-\gamma}$, but the assignment probabilities $\varphi_{ik}$s within are different. Section C.2 illustrates the difference.

## 4. Monte Carlo Estimates

If we have $N_s$ samples $z_i$s from distribution $q(z) = \tilde{q}(z)/Z$ for known normalising constant $Z$, we can use them to estimate $Z_c$ and $Z_d$ in eq. (1). If we only have the unnormalised density, then estimating $\mathcal{L}_\gamma$ requires the normalising factor $Z$. An alternative is to introduce a mixing distribution and use the $\mathcal{L}_\gamma^h$ bound:

$$\mathcal{L}_\gamma^h \approx \frac{1}{1-\gamma} \log \frac{1}{N_s} \sum_i \sum_j p(D|z_{ij})^{1-\gamma}$$
$$- \frac{\gamma}{1-\gamma} \log \frac{1}{N_s} \sum_i \sum_j (q(z_{ij}|u_i)/p(z_{ij}))^{1/\gamma-1},$$

where there are now $N_s'$ samples from $u_i \sim q(u)$ followed by $N_s/N_s'$ samples $z_{ij} \sim q(z|u_i)$. In this setting, $\tilde{q}(z)$ need

not be known explicitly, but we are required to be able to draw the $(u_i, z_i)$s samples and to know the conditional $q(z|u)$ exactly. This particular model for $q(z)$ is the semi-implicit hierarchical construction (Yin & Zhou, 2018).

For $\mathcal{L}_\gamma^{bh}$, we also have $N_s/N_s'$ samples $z_{ik}' \sim r(z|u_i)$. Section B provides more details.

## 5. Experiments

We provide three experiments. The first uses analytical updates to infer the posteriors for a given model, a *variational inference* task; the second and third, Monte Carlo sampling with the reparameterisation trick (Kingma & Welling, 2014) to infer the posteriors and *also* learn the hyperparameters of the model, a *variational learning* task. For $\gamma = 1.0$ we use the standard ELBO implementation directly. We refer the reader to Table 3 in section A as a reminder of the bounds used in the paper and evaluated in this section.

### 5.1. Calibration Study for Mixture Models

We evaluate the quality of the learnt fractional posteriors by examining the calibration diagnostics for a one-dimensional mixture model. We use the Gaussian mixture model (GMM) of Blei et al. (2017, §2.1), where each observation is drawn with white noise (that is, variance $\sigma_{\text{obs}}^2 = 1$) from one of the $K$ components with equal probability, and the component means have independent and identical Gaussian priors. We infer posteriors over component means using variational inference on a set of observations. For a given significance level $\alpha$, actual coverage is the long-term frequency that the $1 - \alpha$ credible interval from a posterior includes the true component mean. The credible interval is calibrated when the actual coverage is $1 - \alpha$. It is known that the posteriors from ELBO are overconfident (Wang & Titterington, 2005).

We use $K = 2$ components centered at $\mu_1 = -2$ and $\mu_2 = 2$, and we compute the empirical coverage $\kappa$ over $5,000$ replicas of $n = 400$ observations (Syring & Martin, 2019, §S2). For each replica, we obtain the approximate fractional posteriors for $\gamma = 0.1$ to $0.9$ in intervals of $0.1$, and the approximate Bayes posterior using ELBO (see section 3.1.3). Using $\alpha = 0.05$, we find that the posteriors from ELBO and $\gamma = 0.9$ are overconfident, that is, $\kappa < 1 - \alpha$; and those from $\gamma \leq 0.8$ are conservative (first set of results in Table 1). Moreover, the interval lengths $\ell$s decrease with $\gamma$. These findings conform to our expectations of fractional posteriors, and they demonstrate that optimising $\mathcal{L}_\gamma$ gives approximate fractional posteriors with the intended properties.

To see the effect of the interval length $\ell$ on $\kappa$, we measure the coverages when, for each replica, the component means are from the Bayes posterior but the component variances are from fractional posteriors with $\gamma = 0.1$, $0.5$ or $0.9$. We find the coverages for such conflated models $C_\gamma$s match that

Table 1: Calibration study of GMM at $\alpha = 0.05$ significance. The empirical coverages $\kappa$ (higher is better) and average interval lengths $\ell$ (shorter better) for each of the $\{\mu_1, \mu_2\}$ means are shown. The last column gives the bounds to $\mathcal{L}_{\text{evd}}$ (higher better). The first set of results is from modelling with different $\gamma$s ($\mathcal{L}_{1.0}$ is ELBO). Results for $\gamma \in \{0.2, 0.4, 0.6, 0.8\}$ are omitted for brevity, but the trend remains. The second set is from conflated models combining the means from ELBO and the variances from $\mathcal{L}_\gamma$. The third set is from calibrations of using the first set of results. The $\gamma$ values for $R_\ell$ and $R_\kappa$ are 0.785 and 0.798.

|  | $\mu_1$ | | $\mu_2$ | | |
|---|---|---|---|---|---|
|  | $\kappa$ | $\ell$ | $\kappa$ | $\ell$ | bound |
| $\mathcal{L}_{0.1}$ | 1.0000 | 0.8515 | 1.0000 | 0.8987 | $-832.5$ |
| $\mathcal{L}_{0.3}$ | 0.9994 | 0.4924 | 0.9988 | 0.5200 | $-833.3$ |
| $\mathcal{L}_{0.5}$ | 0.9876 | 0.3816 | 0.9860 | 0.4029 | $-834.0$ |
| $\mathcal{L}_{0.7}$ | 0.9694 | 0.3225 | 0.9606 | 0.3406 | $-834.4$ |
| $\mathcal{L}_{0.9}$ | 0.9438 | 0.2845 | 0.9334 | 0.3004 | $-834.8$ |
| $\mathcal{L}_{1.0}$ | 0.9278 | 0.2699 | 0.9182 | 0.2850 | $-834.9$ |
| $C_{0.1}$ | 1.0000 | 0.8515 | 1.0000 | 0.8987 | $-839.3$ |
| $C_{0.5}$ | 0.9878 | 0.3816 | 0.9858 | 0.4029 | $-834.5$ |
| $C_{0.9}$ | 0.9436 | 0.2845 | 0.9334 | 0.3004 | $-834.8$ |
| $R_\ell$ | 0.9588 | 0.3046 | 0.9474 | 0.3216 | $-834.6$ |
| $R_\kappa$ | 0.9570 | 0.3021 | 0.9458 | 0.3190 | $-834.6$ |

of the corresponding fractional posteriors in general, but for a minority of replicas changing the variances is insufficient (compare the coverages of $C_\gamma$s to $\mathcal{L}_\gamma$s in Table 1).

Here, we investigate two calibration strategies, one using interval lengths $\ell$s and the other using coverages $\kappa$s. Given our knowledge of the model, sans the locations of the components, we expect $n/K$ observations per component, so their sample mean has variance $K\sigma_{\text{obs}}^2/n$. Combining this with the critical value for $\alpha$ provides an interval length $\ell^*$ that we expect in the ideal case. *For each replica and each component $k$, we perform linear regression on $\gamma$ against $\ell$ using the results from $\mathcal{L}_\gamma$ to predict $\gamma_k^*$ at $\ell^*$. We average the $\gamma_k^*$s to obtain $\gamma^*$ for that replica. The model that optimises $\mathcal{L}_{\gamma^*}$ in then computed, for each replica. We call this $R_\ell$.

The other strategy, called $R_\kappa$, has to be performed *after* computing the $\kappa$s over the replicas. This regresses linearly $\gamma$ against $\kappa$ using the results from $\mathcal{L}_\gamma$ to predict the $\gamma_k^*$ at $1 - \alpha$ coverage, for each component $k$. We then obtain a single $\gamma^*$ by averaging the $\gamma_k^*$s. For each replica, the model that optimises $\mathcal{L}_{\gamma^*}$ is then computed.

In this study, we find both $R_\ell$ and $R_\kappa$ to provide coverages close to $1 - \alpha$ (last set of results in Table 1). For $R_\ell$, the average value of $\gamma^*$ is 0.78, while for $R_\kappa$ the value is 0.80.

In practice, when replications of data sets are not available, bootstrapping can be used (Syring & Martin, 2019). Both $R_\ell$ and $R_\kappa$ are shown to be effective, and which to use in practice will depend primarily on the nature of data collection. Section C.1 gives additional examples; here we note that analysis for $K > 2$ components is complicated by the complex marginal likelihood landscape (Jin et al., 2016).

**Bounds** With current experimental settings, we perform importance sampling using 1,000 samples to estimate the log-evidence to be $-827.2$. So, while $\mathcal{L}_{0.1}$ is the tightest (last column in Table 1) in this scenerio, it is still rather loose. Section C.3 provides more details.

### 5.2. Variational Autoencoder

Variational autoencoder (VAE) (Kingma & Welling, 2014) provides a variational objective to learn the the encoder and decoder neural networks of an autoencoder. Though there are many variations (for example, Tomczak & Welling (2018); Higgins et al. (2017)), we compare with the standard VAE since our aim is to investigate differences with ELBO. VAE is a local latent variable model that separately applies ELBO to each datum. This will be the same for our bounds.

We follow the experimental setup and the neural network models of Ruthotto & Haber (2021) for the gray-scale MNIST dataset (Lecun et al., 1998). The latent space is two-dimensional, so we can inspect the posteriors visually. We make three changes (see section C.4): most significantly we use the *continuous Bernoulli distribution* (Loaiza-Ganem & Cunningham, 2019) as the likelihood function. We use four values of $\gamma$s: 1.0 (ELBO), 0.9, 0.5 and 0.1.

For the posterior family, we use an explicit normal distribution (following Ruthotto & Haber (2021)) and also a semi-implicit distribution. For the latter we use a three-layer neural network for the implicit distribution, similar to that used by Yin & Zhou (2018) (details in section C.4). The choice of posterior family affects only the encoder structure.

In VAE, the log-evidence $\mathcal{L}_{\text{evd}}$ depends on the prior and the likelihood, which in turn depends on the learnt VAE decoder. Hence, *optimising the decoder parameters can be seen as an ML-II procedure* (Wang et al., 2019). Therefore, when comparing the evidence bounds after optimisation, we can only say that the optimised models provide certain guarantees on $\mathcal{L}_{\text{evd}}$, with higher bounds giving better guarantees. For tightness of bounds, see section C.4 (Table 7).

We first look at the effect of $\gamma$ and posterior families, and where $\mathcal{L}_\gamma$ uses the explicit distributions and $\mathcal{L}_\gamma^{\text{h}}$ uses the semi-implicit distributions. While the learning objectives are on the training set, we also examine them on the test set to assess generalisation. We observe smaller $\gamma$s gives higher final evidence bounds (third and fourth columns of Table 2,

Table 2: Average log-evidences (higher better) over data samples, and its breakdown for VAE on MNIST data sets. We give the mean and *three* standard deviations of these averages over ten experimental runs. For Monte Carlo averages, 1,024 samples are used ($32 \times 32$ for semi-implicit posteriors). For $\gamma = 1.0$, the figures are the same under *Test using Objective* and *Test using ELBO*. For the first eight rows, the columns under *Test using ELBO* are *solely* for diagnostics to understand the learnt posteriors using the same metrics: they are not performance measures. For $\mathcal{L}_\gamma^{\mathrm{bh}}$, we show the addition of the KL divergence and Rényi divergence for *Test using Objective*; and for *Test using ELBO* we evaluate the Bayes posterior $r$.

| Objective | $\gamma$ | Train (Total) | Test using Objective | | | Test using ELBO | | |
| --- | --- | --- | --- | --- | --- | --- | --- | --- |
| | | | Total | data | div | Total | data | div |
| $\mathcal{L}_\gamma$ | 1.0 | $1614.3_{\pm 11.0}$ | $1583.2_{\pm 14.6}$ | $1588.5_{\pm 14.5}$ | $5.3_{\pm 0.2}$ | $1583.2_{\pm 14.6}$ | $1588.5_{\pm 14.5}$ | $5.3_{\pm 0.2}$ |
| | 0.9 | $1648.5_{\pm 5.1}$ | $1639.3_{\pm 4.5}$ | $1641.9_{\pm 4.8}$ | $2.6_{\pm 0.4}$ | $1452.6_{\pm 48.0}$ | $1455.0_{\pm 48.0}$ | $2.4_{\pm 0.3}$ |
| | 0.5 | $1675.9_{\pm 5.0}$ | $1672.8_{\pm 5.9}$ | $1674.7_{\pm 6.0}$ | $1.9_{\pm 0.3}$ | $1318.7_{\pm 42.1}$ | $1320.1_{\pm 42.2}$ | $1.4_{\pm 0.2}$ |
| | 0.1 | $1680.1_{\pm 2.9}$ | $1677.2_{\pm 3.4}$ | $1679.5_{\pm 3.4}$ | $2.3_{\pm 0.3}$ | $1322.8_{\pm 49.0}$ | $1324.2_{\pm 49.1}$ | $1.3_{\pm 0.2}$ |
| $\mathcal{L}_\gamma^{\mathrm{h}}$ | 1.0 | $1639.6_{\pm 14.6}$ | $1609.6_{\pm 20.6}$ | $1614.6_{\pm 20.5}$ | $5.0_{\pm 0.3}$ | $1609.7_{\pm 20.6}$ | $1614.6_{\pm 20.5}$ | $5.0_{\pm 0.3}$ |
| | 0.9 | $1657.7_{\pm 6.1}$ | $1647.8_{\pm 5.6}$ | $1651.1_{\pm 5.6}$ | $3.3_{\pm 0.3}$ | $1534.6_{\pm 57.0}$ | $1537.8_{\pm 57.1}$ | $3.2_{\pm 0.3}$ |
| | 0.5 | $1677.4_{\pm 4.1}$ | $1674.4_{\pm 5.0}$ | $1676.4_{\pm 5.0}$ | $2.1_{\pm 0.2}$ | $1366.0_{\pm 62.0}$ | $1367.7_{\pm 62.2}$ | $1.7_{\pm 0.2}$ |
| | 0.1 | $1681.4_{\pm 2.7}$ | $1678.7_{\pm 3.1}$ | $1681.2_{\pm 3.2}$ | $2.5_{\pm 0.2}$ | $1355.6_{\pm 37.7}$ | $1357.1_{\pm 37.8}$ | $1.6_{\pm 0.2}$ |
| $\mathcal{L}_\gamma^{\mathrm{bh}}$ | 0.9 | $1636.2_{\pm 11.5}$ | $1608.8_{\pm 23.3}$ | $1613.4_{\pm 23.0}$ | $4.0_{\pm 0.2} + 0.6_{\pm 0.3}$ | $1607.7_{\pm 23.9}$ | $1613.4_{\pm 23.0}$ | $5.7_{\pm 1.0}$ |
| | 0.5 | $1635.2_{\pm 10.5}$ | $1608.0_{\pm 25.4}$ | $1612.7_{\pm 25.2}$ | $4.3_{\pm 0.2} + 0.4_{\pm 0.2}$ | $1607.4_{\pm 25.8}$ | $1612.7_{\pm 25.3}$ | $5.3_{\pm 0.6}$ |
| | 0.1 | $1635.5_{\pm 12.0}$ | $1607.5_{\pm 16.1}$ | $1612.4_{\pm 16.1}$ | $4.6_{\pm 0.1} + 0.3_{\pm 0.2}$ | $1607.3_{\pm 16.2}$ | $1612.4_{\pm 16.1}$ | $5.1_{\pm 0.2}$ |

first two sets of results), showing that $\mathcal{L}_\gamma$ and $\mathcal{L}_\gamma^{\mathrm{h}}$ can be better than $\mathcal{L}_{\mathrm{ELBO}}$. This implies that using a range of $\gamma$s is useful for model selection, comparison and optimisation. Moreover, $\mathcal{L}_{0.9}$ with the simpler explicit posterior already gives higher bound than $\mathcal{L}_{1.0}^{\mathrm{h}}$ (ELBO) with the semi-implicit posterior (compare their fourth columns), illustrating that $\gamma$ is more impactful than the posterior family. Nonetheless, for the same $\gamma$, the semi-implicit posterior family gives higher evidence bounds (compare the first two sets of results), showing that $\mathcal{L}_\gamma^{\mathrm{h}}$ is a viable approach to learning within the semi-implicit family.

The Rényi and KL divergences generally increase with $\gamma$ (sixth and ninth columns in Table 2). In particular, the trend for KL validates that we are learning fractional posteriors closer to the prior for smaller $\gamma$s. The means of the explicit posterior distributions have also less spread for smaller $\gamma$s (see Fig. 4 in section C); samples from these distributions, which depends on the learnt variances, also demonstrate the same (Fig. 5, section C). This also means that the data is less fitted for smaller $\gamma$s, which is generally shown from the data fit term in ELBO (eighth column in Table 2).

There are more clumps in samples from the semi-explicit posteriors than from the explicit ones (Fig. 5, section C), demonstrating the mixing property of the former. The samples from the implicit posteriors shows that the implicit distribution for $\gamma = 1.0$ (ELBO) are mostly concentrated (Fig. 6a, section C), suggesting frequent degeneracy to the delta distribution, in broad agreement to theory (Yin & Zhou, 2018). In contrast, those for $\gamma < 1$, we find diverse samples

in most cases (Figs. 6b to 6d). This demonstrates that learning with our bounds is a viable alternative to other methods (Yin & Zhou, 2018; Titsias & Ruiz, 2019; Uppal et al., 2023) to prevent collapse of the implicit distributions.

Because of the degeneracy of hierarchical posterior for the ELBO, we have expected to find the results of $\mathcal{L}_{1.0}^{\mathrm{h}}$ to be very similar to that of $\mathcal{L}_{1.0}$. However, this is not the case here. We postulate that the different gradients and the additional implicit samples have led to different learning dynamics and allow $\mathcal{L}_{1.0}^{\mathrm{h}}$ to escape local optimas in the neural network parameter space. The large variances in the train objectives across the ten experimental runs support this.

**Bounds**  For a limited comparison on the tightness of the bounds with respect to a common $\mathcal{L}_{\mathrm{evd}}$, we take a single run of $\mathcal{L}_{\mathrm{ELBO}}$ for the explicit posterior and uses its decoder as the fixed decoder to train the encoders (or posteriors) for $\mathcal{L}_\gamma$. With smaller $\gamma$s, we obtain tighter bounds and posteriors closer to the prior. Details are in section C.4.

### 5.2.1. Comparing $\mathcal{L}_\gamma^{\mathrm{bh}}$ with $\mathcal{L}_\gamma^{\mathrm{h}}$

We examine the joint learning of Bayes posterior $r$ and a fractional posterior $q$ using the bound $\mathcal{L}_\gamma^{\mathrm{bh}}$, where the posteriors are in the same semi-implicit family. We compare with using $\mathcal{L}_\gamma^{\mathrm{h}}$, in both the bounds and the posteriors. The neural network settings are the same as for $\mathcal{L}_\gamma^{\mathrm{h}}$.

The train and test objectives of $\mathcal{L}_\gamma^{\mathrm{bh}}$ do not perform better than those from $\mathcal{L}_\gamma^{\mathrm{h}}$ (and $\mathcal{L}_\gamma$ for $\gamma \neq 1.0$; last three rows in

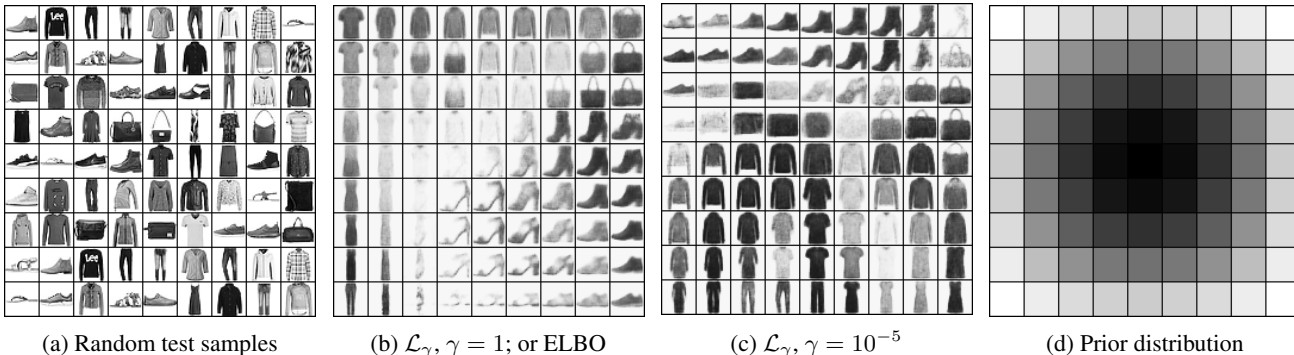

    (a) Random test samples      (b) $\mathcal{L}_\gamma, \gamma = 1$; or ELBO      (c) $\mathcal{L}_\gamma, \gamma = 10^{-5}$      (d) Prior distribution

Figure 1: We train VAEs on the Fashion-MNIST dataset using $\mathcal{L}_\gamma$ for different $\gamma$s. We obtain mean images from the decoding latent variables that are systematically sampled by coordinate-wise inverse-CDF (standard normal) transform from a unit square. Fig. b shows the images using the Bayes posterior (learnt with ELBO), and Fig. c shows those using a fractional posterior very close to the prior. The last image is the heat map of the corresponding prior densities.

Table 2), even though there are more parameters — those for the Bayes posterior $r$. In particular, they do not perform better than $\mathcal{L}_{1.0}^{\mathrm{h}}$, as would be suggested by Lemma A.4; but we qualify that the decoders and hence the probabilistic models are probably different.

We find $\mathrm{KL}[r\|q]$ to be large, and it increases with smaller $\gamma$, as expected (first summand in the sixth column in Table 2). When we evaluate the Bayes posteriors $r$s with ELBO, we find that they are competitive with those obtained by directly optimising $\mathcal{L}_{1.0}^{\mathrm{h}}$ (seventh column).

Comparing the Rényi divergences of the fractional posteriors learnt with $\mathcal{L}_\gamma^{\mathrm{bh}}$ to those learnt with $\mathcal{L}_\gamma^{\mathrm{h}}$ (sixth column in Table 2), we find those learnt with $\mathcal{L}_\gamma^{\mathrm{bh}}$ significantly closer to the prior. This shows that the fractional posteriors from $\mathcal{L}_\gamma^{\mathrm{bh}}$ are constrained significantly by the Bayes posteriors when learnt jointly (see third paragraph in section 2.3.2).

### 5.3. Improving VAE Decoder via Fractional Posteriors

The decoder of VAE is learnt with latent samples from the encoder. An encoder from a fractional posterior gives samples closer to the prior than the Bayes posterior. Hence, when we generate images from the VAE decoder using samples from the prior—that is, without using the encoder that need an input data—we expect the decoder learnt with a smaller $\gamma$ to provide better images.

We illustrate this with the Fashion-MNIST dataset (Xiao et al., 2017), training with $\mathcal{L}_\gamma$ for $\gamma$ taking values 1.0 (for Bayes posterior), $10^{-1}$, $10^{-3}$ and $10^{-5}$ (for fractional posterior close to prior). Using latent samples from the prior, the decoder trained with the Bayes posterior provides images that are of lower quality then the decoder trained with the fractional posterior with $\gamma = 10^{-5}$ (Fig. 1; and Fig. 7 in section C.5). To quantify, we generate 10,000 images from each trained decoder and measure their Fréchet inception

distances (FIDs, Heusel et al. 2017; Seitzer 2020) to the test set: with decreasing $\gamma$, the distances are 83.5, 69.5, 67.8 and 68.8 (smaller is better). This shows that fractional posteriors can train better decoders for generative modelling.

The $\beta$-VAE objective (Higgins et al., 2017) with the appropriate parameters also gives fractional posteriors. However, this objective can be unstable during optimisation, especially when we seek fractional posteriors very close to the prior. For the same fractional posteriors with $\gamma$ set to $10^{-1}$, $10^{-3}$ and $10^{-5}$, we obtain FIDs 77.3, 334.7 and 342.3. Section C.5 provides the details.

## 6. Related Work

*Generalised variational inference* (Knoblauch et al., 2022) provides an optimisation framework that generalises ELBO. Being generic, one need to concretise the individual terms before applying to specific cases. One example is $\beta$-VAE (Higgins et al., 2017) for learning disentangled representations in variational autoencoders (VAEs): it weighs the divergence term more heavily. By construction, the $\beta$-VAE bound is provably not tighter than ELBO, and optimising it gives a fractional posterior. The importance weighted ELBO is also not tighter than ELBO (Domke & Sheldon, 2018, eq. 8). In contrast, the variational Rényi bound (Li & Turner, 2016) can be tighter than ELBO, and optimising it gives the Bayes posterior. This paper provides a bound that can be better than ELBO, especially for simpler assumed family of distributions, and optimising it gives a fractional posterior. We show this with the calibration and VAE study.

In a standard VAE, the decoder is trained with samples from the posteriors encoded using the training data, but these are unavailable for pure generative tasks. Current approaches overcome this by learning a prior that is accessible during generation, with the objective for matching the prior to

the posteriors (Makhzani et al., 2016; Tomczak & Welling, 2018; Tran et al., 2021). The alternative is to train the decoder via a distribution close to the prior. While the $\beta$-VAE implies such a distribution, its looser bound suggests that the decoder parameters may be learnt suboptimally. Our approach uses fractional posteriors and gives bounds higher than ELBO empirically.

Variational inference can be seen as *intentional* model misspecification (Chen et al., 2018). Fractional posteriors is one approach to overcome misspecification (Grünwald & van Ommen, 2017). Such posteriors can be obtained by sampling with down-weighted likelihood; or one can adjust the scale parameter of the Bayes posterior (Syring & Martin, 2019). Alternatively, one can optimise the $\beta$-VAE objective (Alquier et al., 2016; Higgins et al., 2017). We have provided an alternative variational approach to approximate fractional posteriors, and we have demonstrated calibration using them within regression procedures. More complex calibration procedures (Grünwald & van Ommen, 2017; Syring & Martin, 2019) can be explored.

## 7. Discussions and Limitations

**Misspecification**  If the prior and likelihood are correctly given, and if exact inference is possible, then in principle a Bayesian only needs to compute the exact Bayes posterior. This seldom happens in practice (Faden & Rausser, 1976). If either the prior or likelihood or both are misspecified, then post-Bayesianism efforts, such as generalised Bayes (Knoblauch et al., 2022), robust Bayesian (Miller & Dunson, 2019) and PAC-Bayes (Masegosa, 2020; Morningstar et al., 2022), seek to ameliorate the situation. This paper does not directly address the goals of post-Bayesianism. It has not evaluated when either the likelihood or the prior is misspecified. Although those for MNIST and Fashion-MNIST are most probably misspecified, we have not compared to when they are not. We rely on the works of others to address such goals. Nonetheless, we make a two connections here.

First, using fractional posteriors is a proposed solution for misspecification (Grünwald & van Ommen, 2017; Bhattacharya et al., 2019; Medina et al., 2022), and our objective does this naturally through a lower bound on the log-evidence. While the objective is not as intuitive as, say, the $\beta$-VAE objective, a lower bound like $\mathcal{L}_\gamma$ that can be tighter than ELBO can help in selecting or optimising appropriately parameterised priors in an empirical Bayes manner (Berger & Berliner, 1986). Second, our objective uses the Rényi divergence to quantify the closeness of the prior to the posterior, and this divergence is well-behaved for robustness to prior misspecification (Knoblauch et al., 2022, §5.2.1).

When exact inference is not possible, we can use approximate inference by way of variational optimisation, which is the main alternative to Monte Carlo approaches. When the assumed variational family does not include the exact Bayes posterior, some—but not all—also consider this misspecification (Chen et al., 2018; Knoblauch et al., 2022). This paper addresses this by expanding the possibility afforded by the conventional ELBO, so that we may also have approximate fractional posteriors as the optimal solutions. At this point, there is no single recipe to select $\gamma$ (section 2.2); we opine that this should be application dependent. For example in section 5.1, the best $\gamma$s are selected for calibration and not for the tightness of the corresponding bounds.

**Posterior collapse**  With small $\gamma$, we might seem to be encouraging posterior collapse (Wang et al., 2021). However, Fig. 4d for $\gamma = 0.1$ demonstrates that while the fractional posteriors as a whole aggregate towards the prior, the posterior for every data point is different. This topic demands more investigation and discussion than possible here.

**Limitations**  We identify three limitations with $\mathcal{L}_\gamma$. First, the conventional ELBO using the KL divergence is usually more mathematically elegant and convenient. This is because it can convert a log-sum (or log-integral) to a sum-log (or integral-log), and a sum or integral is neccessary for marginalisation. In particular, the variational inference for the mixture model must rely on the ELBO at some point (section 3.1.3). Second, the Rényi divergence is finite in less cases than the KL (Gil et al., 2013). It may be necessary to consider this when optimising the parameters of the approximate posteriors, though we have not needed to do this for the presented experiments. Third, when using Monte Carlo estimates, more than one sample is required to be effective (section B.1). This can be unrealistic for huge datasets.

**Power posteriors**  This paper focuses on estimating an approximate fractional posteriors with the lower bound $\mathcal{L}_\gamma$ on the log-evidence. For power posteriors with $\gamma > 1$, we have upper bounds (Lemma A.1). Similarly, there is no fixed criterion to select such $\gamma$s (section 2.2). Estimating these posteriors involves minimising the upper bounds, but machine learning applications will typically require maximising the minimised upper bounds; executing this is probably more involved than what is done in this paper.

## 8. Conclusions

We have presented a novel one-parameter variational lower bound for the evidence. Maximising the bound within an assumed family of distributions estimates approximate fractional posteriors. We have given analytical updates for approximate inference in two intractable models. Empirical results for calibration and VAE show the utility of our approach. For the Fashion-MNIST dataset, VAE decoders learnt with our approach can generate better images.

## Acknowledgements

We thank Xuesong Wang and Rafael Oliveira for their inputs, and the reviewers for their constructive comments. This research was conducted while Chai was visiting CSIRO's Data61, and the Machine Learning and Data Science Unit in the Okinawa Institute of Science and Technology (OIST).

## Impact Statement

This paper presents work whose goal is to advance the field of Machine Learning. There are many potential societal consequences of our work, none which we feel must be specifically highlighted here.

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

# A. Proofs

This section collects the proofs for the main paper. For reference, Table 3 lists the bounds used in this paper.

**Lemma A.1.** $\mathcal{L}_{\text{evd}}$ *is upper bounded the same expression as eq. (1), but with* $\gamma > 1$.

*Proof.* Similar to the proof for $\gamma \in (0, 1)$, but we use the *reverse* Hölder's inequality in the manner of $\mathbb{E}\left[|X|^{1/\beta'}\right]^{\beta'} \leq \mathbb{E}[|XY|] / \mathbb{E}\left[|Y|^{1/\gamma'}\right]^{\gamma'}$, where $\beta' + \gamma' = 1$ and $\beta' < 0$, with the same expressions for $\mathbb{E}[\cdot]$, $X$ and $Y$. Set $\gamma \equiv \gamma' = 1 - \beta' > 0$. $\square$

**Lemma A.2.** $\mathcal{L}_{\text{evd}}$ *is upper bounded the same expression as eq. (1), but with* $\gamma < 0$.

*Proof.* Similar to the proof for Lemma A.1, but now we use $\beta' > 1$, so that $\gamma \equiv \gamma' = 1 - \beta' < 0$. $\square$

**Lemma A.3.** $\lim_{\gamma \to 1} \mathcal{L}_\gamma = \mathcal{L}_{\text{ELBO}}$.

*Proof.* The data term in $\mathcal{L}_\gamma$ converges to the expectation of the log likelihood under approximate posterior $q$ when we apply the L'Hôpital's rule:

$$
\begin{aligned}
\lim_{\gamma \to 1} \frac{\log \int \tilde{q}(z) \, p(D|z)^{1-\gamma} \mathrm{d}z}{1 - \gamma} &= \lim_{\gamma \to 1} \frac{\int \tilde{q}(z) \, p(D|z)^{1-\gamma} \log p(D|z) \mathrm{d}z}{\int \tilde{q}(z) \, p(D|z)^{1-\gamma} \mathrm{d}z} \\
&= \frac{\int \tilde{q}(z) \log p(D|z) \mathrm{d}z}{\int \tilde{q}(z) \mathrm{d}z} \\
&= \int q(z) \log p(D|z) \mathrm{d}z
\end{aligned}
$$

Moreover, as $\gamma \to 1$, the Rényi divergence converges to the KL divergence (van Erven & Harremos, 2014). $\square$

**Lemma A.4.** *For a fix approximate Bayes posterior,* $\mathcal{L}_\gamma^{\text{b}} \leq \mathcal{L}_{\text{ELBO}}$.

*Proof.* For a fixed $r(z)$, $\mathcal{L}_\gamma^{\text{b}}$ is optimal at $\tilde{q}^*(z) \propto r(z)^\gamma p(z)^{1-\gamma}$. Substituting $\tilde{q}^*(z)$ into $\mathcal{L}_\gamma^{\text{b}}$ recovers ELBO. Hence, $\mathcal{L}_\gamma^{\text{b}}$ is upper bounded by ELBO. $\square$

## A.1. Variational Derivation of Saddle Point of $\mathcal{L}_\gamma$

To obtain the functional derivative $\partial \mathcal{L}_\gamma / \partial \tilde{q}$, we introduce scalar $h$ and function $\eta(z)$ and consider

$$
\begin{aligned}
\left.\frac{\mathrm{d}\mathcal{L}_\gamma(\tilde{q} + h\eta)}{\mathrm{d}h}\right|_{h=0} &= \left(\frac{1}{1-\gamma} \frac{\int \eta(z) \, p(D|z)^{1-\gamma} \mathrm{d}z}{\int (\tilde{q}(z) + h\eta(z)) \, p(D|z)^{1-\gamma} \mathrm{d}z} - \frac{1}{1-\gamma} \frac{\int \eta(z) \, (\tilde{q}(z) + h\eta(z))^{1/\gamma-1} p(z)^{1-1/\gamma} \mathrm{d}z}{\int (\tilde{q}(z) + h\eta(z))^{1/\gamma} p(z)^{1-1/\gamma} \mathrm{d}z}\right)_{h=0} \\
&= \frac{1}{1-\gamma} \frac{\int \eta(z) \, p(D|z)^{1-\gamma} \mathrm{d}z}{\int \tilde{q}(z) \, p(D|z)^{1-\gamma} \mathrm{d}z} - \frac{1}{1-\gamma} \frac{\int \eta(z) \, \tilde{q}(z)^{1/\gamma-1} p(z)^{1-1/\gamma} \mathrm{d}z}{\int \tilde{q}(z)^{1/\gamma} p(z)^{1-1/\gamma} \mathrm{d}z} \\
&= \int \frac{1}{1-\gamma} \left(\frac{p(D|z)^{1-\gamma}}{\int \tilde{q}(z') \, p(D|z')^{1-\gamma} \mathrm{d}z'} - \frac{\tilde{q}(z)^{1/\gamma-1} p(z)^{1-1/\gamma}}{\int \tilde{q}(z')^{1/\gamma} p(z')^{1-1/\gamma} \mathrm{d}z'}\right) \eta(z) \mathrm{d}z,
\end{aligned}
$$

where the integrand sans $\eta(z)$ is the required derivative. Equating to zero give

$$
\tilde{q}(z) = p(D|z)^\gamma p(z) / \tilde{Z} \qquad\qquad \tilde{Z} = \left(\frac{\int \tilde{q}(z) \, p(D|z)^{1-\gamma} \mathrm{d}z}{\int \tilde{q}(z)^{1/\gamma} p(z)^{1-1/\gamma} \mathrm{d}z}\right)^{\gamma/(1-\gamma)}
$$

Substituting $\tilde{q}(z)$ into the RHS of $\tilde{Z}$ gives

$$
\tilde{Z} = \left(\frac{\int p(z) \, p(D|z) \mathrm{d}z / \tilde{Z}}{\int p(D|z) p(z) \mathrm{d}z / \tilde{Z}^{1/\gamma}}\right)^{\gamma/(1-\gamma)} = \tilde{Z},
$$

Table 3: List of lower bounds. Some are expressed differently from the main paper to ease comparison among the bounds. The $\beta$-VAE objective is the weighted KL divergence (Knoblauch et al., 2022, §B.3.1).

| Description | Notation | Expression |
|---|---|---|
| Log-evidence | $\mathcal{L}_{\mathrm{evd}}$ | $\log p(D) = \log \int p(z)p(D\vert z)\mathrm{d}z$ |
| ELBO (evidence lower bound) | $\mathcal{L}_{\mathrm{ELBO}}$ | $\int q(z) \log p(D\vert z)\mathrm{d}z - \int q(z)\log \frac{q(z)}{p(z)}\mathrm{d}z$ |
| Weighted KL divergence ($\beta$-VAE objective) | $\mathcal{L}_{\beta}^{\beta}$ | $\int q(z) \log p(D\vert z)\mathrm{d}z - \beta \int q(z)\log \frac{q(z)}{p(z)}\mathrm{d}z$ |
| Variational Rényi bound | $\mathcal{L}_{\alpha}^{\mathrm{R}}$ | $\frac{1}{1-\alpha} \log \int q(z) \left( \frac{p(D\vert z)\,p(z)}{q(z)} \right)^{1-\alpha} \mathrm{d}z$ |
| Our primary bound | $\mathcal{L}_{\gamma}$ | $\frac{1}{1-\gamma} \log \int \tilde{q}(z)p(D\vert z)^{1-\gamma}\mathrm{d}z - \frac{\gamma}{1-\gamma} \log \int \tilde{q}(z) \left( \frac{\tilde{q}(z)}{p(z)} \right)^{1/\gamma-1} \mathrm{d}z$ |
| • with hierarchical fractional posterior | $\mathcal{L}_{\gamma}^{\mathrm{h}}$ | $\frac{1}{1-\gamma} \log \iint \tilde{q}(z\vert u)q(u)p(D\vert z)^{1-\gamma}\mathrm{d}z\mathrm{d}u$ $- \frac{\gamma}{1-\gamma} \log \iint \tilde{q}(z\vert u)q(u) \left( \frac{\tilde{q}(z\vert u)}{p(z)} \right)^{1/\gamma-1} \mathrm{d}z\,\mathrm{d}u$ |
| • with Bayes posterior | $\mathcal{L}_{\gamma}^{\mathrm{b}}$ | $\int r(z) \log p(D\vert z)\mathrm{d}z - \frac{1}{1-\gamma} \int r(z)\log \frac{r(z)}{\tilde{q}(z)}\mathrm{d}z$ $- \frac{\gamma}{1-\gamma} \log \int \tilde{q}(z) \left( \frac{\tilde{q}(z)}{p(z)} \right)^{1/\gamma-1} \mathrm{d}z$ |
| • with hierarchical fractional and Bayes posteriors | $\mathcal{L}_{\gamma}^{\mathrm{bh}}$ | $\iint r(z\vert u)q(u) \log p(D\vert z) \, \mathrm{d}z\mathrm{d}u$ $- \frac{1}{1-\gamma} \iint r(z\vert u)q(u) \log \frac{r(z\vert u)}{\tilde{q}(z\vert u)}\mathrm{d}z\mathrm{d}u$ $- \frac{\gamma}{1-\gamma} \log \iint \tilde{q}(z\vert u)q(u) \left( \frac{\tilde{q}(z\vert u)}{p(z)} \right)^{1/\gamma-1} \mathrm{d}z\mathrm{d}u$ |
| • with hierarchical fractional and Bayes posteriors (alternative bound; see section A.4) | $\mathcal{L}_{\gamma}^{\mathrm{bh}}$-alt | $\iint r(z\vert u)q(u) \log p(D\vert z) \, \mathrm{d}z\mathrm{d}u$ $- \frac{1}{1-\gamma} \iiint r(z\vert u)q(u)q(u') \log \frac{r(z\vert u)}{\tilde{q}(z\vert u')}\mathrm{d}z\mathrm{d}u\mathrm{d}u'$ $- \frac{\gamma}{1-\gamma} \log \iint \tilde{q}(z\vert u)q(u) \left( \frac{\tilde{q}(z\vert u)}{p(z)} \right)^{1/\gamma-1} \mathrm{d}z\mathrm{d}u$ |

which is self-consistent. The solution is true for any $\tilde{Z}$, so we *may* choose $\tilde{Z}$ as the normalising constant. Because of this, and because the optimal solution is already non-negative, it is not necessary to introduction Lagrange multipliers for constrained optimisation.

## A.2. Gap between Log-evidence and Variational Rényi Bound

$$
\begin{aligned}
\log p(D) - \frac{1}{1-\alpha} \log \int q(z) \left(p(D,z)/q(z)\right)^{1-\alpha} \mathrm{d}z &= \frac{1}{1-\alpha} \log \frac{p(D)^{1-\alpha}}{\int q(z) \left(p(D,z)/q(z)\right)^{1-\alpha} \mathrm{d}z} \\
&= \frac{1}{\alpha-1} \log \int \frac{q(z) \left(p(D,z)/q(z)\right)^{1-\alpha}}{p(D)^{1-\alpha}} \mathrm{d}z \\
&= \frac{1}{\alpha-1} \log \int q(z)^{\alpha} p(z|D)^{1-\alpha} \mathrm{d}z,
\end{aligned}
$$

## A.3. Divergences

Following the definition of $\mathcal{L}_{\mathrm{evd}}$ and its lower bound $\mathcal{L}_\gamma$, we may define a divergence $\mathrm{D}_\gamma^{\mathrm{frac}}$ from distribution $p_2$ to distribution $p_1$ with respect to an underlying distribution $p_0$:

$$
\mathrm{D}_\gamma^{\mathrm{frac}} [p_2 \| p_1] \overset{\mathrm{def}}{=} \mathcal{L}_{\mathrm{evd}} - \mathcal{L}_\gamma.
$$

Here, $p_0$ participates as the prior, $p_1(z) \propto \ell(z)^\gamma p_0(z)$ as the target fractional posterior with likelihood $\ell(z)$, and $p_2 \equiv q$ as the approximating posterior. By definition, the divergence is non-negative because $\mathcal{L}_\gamma$ is a lower bound, and the divergence is zero when $p_2 = p_1$ because $\mathcal{L}_\gamma$ is tight.

Let $Z$ be the normalising constant of $p_1$. We have $\ell(z) = Z^{1/\gamma}(p_1(z)/p_0(z))^{1/\gamma}$. By substitution and simplification,

$$
\begin{aligned}
\mathcal{L}_{\mathrm{evd}} &= \frac{1}{\gamma} \log Z + \log \int p_1(z)^{1/\gamma} p_0(z)^{1-1/\gamma} \mathrm{d}z \\
\mathcal{L}_\gamma &= \frac{1}{\gamma} \log Z + \frac{1}{1-\gamma} \log \int p_2(z) p_1(z)^{1/\gamma-1} p_0(z)^{1-1/\gamma} \mathrm{d}z - \frac{\gamma}{1-\gamma} \log \int p_2(z)^{1/\gamma} p_0(z)^{1-1/\gamma} \mathrm{d}z.
\end{aligned}
$$

Therefore,

$$
\begin{aligned}
\mathrm{D}_\gamma^{\mathrm{frac}} [p_2 \| p_1] = {}& \log \int p_1(z)^{1/\gamma} p_0(z)^{1-1/\gamma} \mathrm{d}z - \frac{1}{1-\gamma} \log \int p_2(z) p_1(z)^{1/\gamma-1} p_0(z)^{1-1/\gamma} \mathrm{d}z \\
& + \frac{\gamma}{1-\gamma} \log \int p_2(z)^{1/\gamma} p_0(z)^{1-1/\gamma} \mathrm{d}z,
\end{aligned}
$$

and $Z$ is not required.

We may also obtain a divergence without the notion of fractional posterior. Continuing from above, let $\tilde{p}_i(z) \overset{\mathrm{def}}{=} p_i(z)^{1/\gamma} p_0(z)^{1-1/\gamma}/Z_i$, for $i = 1,2$ and where $Z_i$s are normalising constants. Then, by substituting into and simplifying expressions in the right side of the above equation, we have

$$
\frac{1}{\gamma-1} \log \int \tilde{p}_2(z)^\gamma \tilde{p}_1(z)^{1-\gamma} \mathrm{d}z,
$$

which is the Rényi divergence $\mathrm{D}_\gamma[\tilde{p}_2 \| \tilde{p}_1]$.

Observe that $\tilde{p}_1(z) \propto \ell(z) p_0(z)$ is the Bayes posterior. So, in minimising the Rényi divergence, we obtain $\tilde{p}_2$ as an approximate Bayes posterior. We recover an approximate fractional posterior $p_2$ by using the definition of $\tilde{p}_2$, where the prior $p_0$ is needed. Therefore, if we are to change the subject of optimisation from the fractional posterior to the Bayes posterior, we recover the variational Rényi bound (Li & Turner, 2016).

Throughout this section, the precise definition of $\mathcal{L}_\gamma$ has allowed normalising constants to be cancelled. Also, we can derive $\mathcal{L}_\gamma$ as a lower bound retrospectively by reading this section in reverse.

**A.4. Derivations for $\mathcal{L}_\gamma^{\text{bh}}$**

The data term is because $p(D|z)$ is not dependent on $u$. The Rényi divergence term follows from eq. (2). We address the KL divergence term below.

We use the convexity of $\text{KL}[p\|q]$ in the pair $(p, q)$. This provides $-\int r(z) \log(r(z)/q(z)) \mathrm{d}z \geq -\int \left(\int r(z|u) \log(r(z|u)/q(z|u)) \mathrm{d}z\right) q(u) \mathrm{d}u$ for the KL divergence term in $\mathcal{L}^{\text{bh}}$. The overall bound requires a double integral because we have a hierarchical construction, involving random variables $u$ and $z$ given $u$.

An alternative derivation gives $\mathcal{L}_\gamma^{\text{bh}}$-alt in the last row in Table 3. The function $-x \log x$ is concave, so we have $-\int r(z) \log r(z) \mathrm{d}z \geq -\int \left(\int r(z|u) \log r(z|u) \mathrm{d}z\right) q(u) \mathrm{d}u$. Similarly, $\log x$ is concave, so we also have $\int r(z) \log \tilde{q}(z) \mathrm{d}z \geq \int r(z) \left(\int q(u') \log \tilde{q}(z|u') \mathrm{d}z\right) \mathrm{d}u'$. Introducing $u'$ to the entropy term and $u$ to the negative cross-entropy term and then summing the two gives an alternative KL divergence term in $\mathcal{L}^{\text{bh}}$. The triple integral in the KL term comes from the *independently* mixing of $r(z|u)$ and $\tilde{q}(z|u)$ with the same distribution $q(u)$.

**A.5. Non-degeneracy of the Implicit Distributions within the Semi-implicit Distributions**

This section shows the existence of non-degenerate implicit distributions when optimising for $\mathcal{L}_\gamma^{\text{h}}$, $\mathcal{L}_\gamma^{\text{bh}}$ and $\mathcal{L}_\gamma^{\text{h}}$-alt. A common theme is that the Rényi divergence is expressed as a log-integral rather than an integral-log, so the optimsation for the implicit distribution cannot be factored out.

A.5.1. FOR FRACTIONAL POSTERIORS USING $\mathcal{L}_\gamma^{\text{h}}$

For this section, let $f(u) \stackrel{\text{def}}{=} \int q(z|u) p(D|z)^{1-\gamma} \mathrm{d}z$ and $g(u) \stackrel{\text{def}}{=} \int (q(z|u)/p(z))^{1/\gamma - 1} q(z|u) \mathrm{d}z$, so that $\mathcal{L}_\gamma^{\text{h}}(q) = (\log \int f(u) q(u) \mathrm{d}u)/(1 - \gamma) - (\log \int g(u) q(u) \mathrm{d}u) \gamma/(1 - \gamma)$, where $q$ is just for the distribution of $u$. We introduce scalar $h$ and function $\eta(u)$. The functional derivative $\partial \mathcal{L}_\gamma^{\text{h}}/\partial \log q$ is the integrand sans $\eta(u)$ of the expression

$$\frac{\mathrm{d}\mathcal{L}_\gamma^{\text{h}}(\log q + h\eta)}{\mathrm{d}h}\bigg|_{h=0} = \int \left(\frac{1}{1 - \gamma} \frac{f(u)}{\int f(u') q(u') \mathrm{d}u'} - \frac{\gamma}{1 - \gamma} \frac{g(u)}{\int g(u') q(u') \mathrm{d}u'}\right) q(u) \eta(u) \mathrm{d}u$$

Together with the normalisation constraint which introduces a Lagrange multiplier $\lambda$, we require

$$\left(\frac{1}{1 - \gamma} \frac{f(u)}{\int f(u') q(u') \mathrm{d}u'} - \frac{\gamma}{1 - \gamma} \frac{g(u)}{\int g(u') q(u') \mathrm{d}u'} + \lambda\right) q(u) = 0.$$

Integrating with respect to $u$ yield $1 + \lambda = 0$, so we have

$$\left(\frac{1}{1 - \gamma} \frac{f(u)}{\int f(u') q(u') \mathrm{d}u'} - \frac{\gamma}{1 - \gamma} \frac{g(u)}{\int g(u') q(u') \mathrm{d}u'} - 1\right) q(u) = 0. \tag{3}$$

This means that $q(u)$ will collapse to zero if the left term is not zero. Though it is not necessary that $q(u)$ degenerates to a delta distribution, a delta distribution satisfies the above constraint readily.

For further illustration, consider $q(u)$ supported at only two locations $u_1$ and $u_2$. Using $q_i$, $f_i$ and $g_i$ to denote evaluations of $f$ and $g$ at these locations, we have

$$q_1 = \frac{(1 - \gamma) f_2 g_2 - f_1 g_2 + \gamma f_2 g_1}{(1 - \gamma)(f_1 - f_2)(g_1 - g_2)} \qquad q_2 = \frac{(1 - \gamma) f_1 g_1 - f_2 g_1 + \gamma f_1 g_2}{(1 - \gamma)(f_1 - f_2)(g_1 - g_2)},$$

which is satisfiable with different values of $f_i$s and $g_i$s.

A.5.2. FOR BAYES AND FRACTIONAL POSTERIORS USING $\mathcal{L}_\gamma^{\text{bh}}$

For this section, let

$$f(u) \stackrel{\text{def}}{=} \int r(z|u) \log p(D|z) \mathrm{d}z \qquad g(u) \stackrel{\text{def}}{=} \int (q(z|u)/p(z))^{1/\gamma - 1} q(z|u) \mathrm{d}z$$

$$h(u) \stackrel{\text{def}}{=} \int r(z|u) \log r(z|u) \mathrm{d}z \qquad d(u) \stackrel{\text{def}}{=} \int r(z|u) \log q(z|u) \mathrm{d}z,$$

so that

$$\mathcal{L}_\gamma^{\text{bh}}(q) = \int f(u)q(u)\mathrm{d}u - \frac{1}{1-\gamma} \int h(u)q(u)\mathrm{d}u + \frac{1}{1-\gamma} \int d(u)q(u)\mathrm{d}u - \frac{\gamma}{1-\gamma} \log \int g(u)q(u)\mathrm{d}u,$$

where $q$ is just for the distribution of $u$. Taking derivative with respect to $\log q(u)$ and imposing normalisation constraint with the Lagrange multiplier $\lambda$, we require

$$\left( f(u) - \frac{1}{1-\gamma}h(u) + \frac{1}{1-\gamma}d(u) - \frac{\gamma}{1-\gamma}\frac{g(u)}{\int g(u')q(u')\mathrm{d}u'} - \lambda \right) q(u) = 0. \tag{4}$$

Integrating with respect to $u$ fix

$$\lambda = \int f(u)q(u)\mathrm{d}u - \frac{1}{1-\gamma} \int h(u)q(u)\mathrm{d}u + \frac{1}{1-\gamma} \int d(u)q(u)\mathrm{d}u - \frac{\gamma}{1-\gamma}.$$

A delta distribution satisfies eq. (4) readily. However, other solutions are also possible in general. As an example, consider $q(u)$ supported at only two locations $u_1$ and $u_2$, and let $\Delta f \overset{\text{def}}{=} f(u_1) - f(u_2)$, $\Delta h \overset{\text{def}}{=} h(u_1) - h(u_2)$, $\Delta d \overset{\text{def}}{=} d(u_1) - d(u_2)$, and $\Delta g \overset{\text{def}}{=} g(u_1) - g(u_2)$. In these settings, and using $q(u_1) + q(u_2) = 1$, eq. (4) may be written as

$$\left( \Delta f - \frac{1}{1-\gamma}\Delta h + \frac{1}{1-\gamma}\Delta d - \frac{\gamma}{1-\gamma}\frac{\Delta g}{g(u_1)q(u_1) + g(u_2) + q(u_2)} \right) q(u_1)q(u_2) = 0$$

Since $q(u_1) \neq 0$ and $q(u_2) \neq 0$, the first term must be zero. This can be expressed as

$$g(u_1)q(u_1) + g(u_2)q(u_2) = \frac{\gamma\Delta g}{(1-\gamma)\Delta f - \Delta h + \Delta d}.$$

Using $q(u_2) = 1 - q(u_1)$, the explicit expression for $q(u_1)$ is

$$q(u_1) = \frac{\gamma}{(1-\gamma)\Delta f - \Delta h + \Delta d} - g(u_2).$$

### A.5.3. FOR BAYES AND FRACTIONAL POSTERIORS USING $\mathcal{L}_\gamma^{\text{bh}}$-ALT

We define $f$, $g$ and $h$ as for $\mathcal{L}_\gamma^{\text{bh}}$ in section A.5.2, but we now have $d(u, u') \overset{\text{def}}{=} \int r(z|u) \log q(z|u')\mathrm{d}z$, so that

$$\mathcal{L}_\gamma^{\text{bh}}\text{-alt}(q) = \int f(u)q(u)\mathrm{d}u - \frac{1}{1-\gamma} \int h(u)q(u)\mathrm{d}u + \frac{1}{1-\gamma} \iint d(u, u')q(u)q(u')\mathrm{d}u'\mathrm{d}u$$
$$- \frac{\gamma}{1-\gamma} \log \int g(u)q(u)\mathrm{d}u,$$

where $q$ is just for the distribution of $u$. Taking derivative with respect to $\log q(u)$ and imposing normalisation constraint with the Lagrange multiplier $\lambda$, we require

$$\left( f(u) - \frac{1}{1-\gamma}h(u) + \frac{1}{1-\gamma}\int (d(u,u') + d(u',u))q(u')\mathrm{d}u' - \frac{\gamma}{1-\gamma}\frac{g(u)}{\int g(u')q(u')\mathrm{d}u'} - \lambda \right) q(u) = 0. \tag{5}$$

Integrating with respect to $u$ fix

$$\lambda = \int f(u)q(u)\mathrm{d}u - \frac{1}{1-\gamma} \int h(u)q(u)\mathrm{d}u + \frac{2}{1-\gamma} \iint d(u,u')q(u)q(u')\mathrm{d}u'\mathrm{d}u - \frac{\gamma}{1-\gamma}.$$

A delta distribution satisfies eq. (5) readily. However, other solutions are also possible in general. As an example, consider $q(u)$ supported at only two locations $u_1$ and $u_2$, and let $\Delta f \overset{\text{def}}{=} f(u_1) - f(u_2)$, $\Delta h \overset{\text{def}}{=} h(u_1) - h(u_2)$, $\Delta d \overset{\text{def}}{=} \int (d(u_1, u') + d(u', u_1))q(u')\mathrm{d}u' - \int (d(u_2, u') + d(u', u_2))q(u')\mathrm{d}u'$, and $\Delta g \overset{\text{def}}{=} g(u_1) - g(u_2)$. We proceed as the example for section A.5.2 under these definitions.

## A.6. Gradients and Expectations for the Multinomial Example

The gradients with respect to the parameters are

$$\frac{\partial \log \tilde{q}(\boldsymbol{z})}{\partial \mu_c} = \frac{z_c - \mu_c}{\sigma_c^2} \qquad\qquad \frac{\partial \log \tilde{q}(\boldsymbol{z})}{\partial \sigma_c^2} = -\frac{1}{2\sigma_c^2} + \frac{(z_c - \mu_c)^2}{2\sigma_c^4}$$

and the required expectations are

$$\mathbb{E}_{q_c}[z_c] = m_c + \rho_c s_c^2 \qquad\qquad \mathbb{E}_{q_c}\left[z_c^2\right] = s_c^2 + m_c^2 + s_c^2(s_c^2 + 2m_c)\rho_c$$

$$\mathbb{E}_{q_d}[z_c] = \mu_c + \sigma_c^2 \, n_c/(n+1) \qquad\qquad \mathbb{E}_{q_d}\left[z_c^2\right] = \sigma_c^2 + \left(\mu_c + \sigma_c^2 \, n_c/(n+1)\right)^2,$$

where

$$\rho_c \stackrel{\text{def}}{=} \frac{\exp(m_c + s_c^2/2)}{\sum_{c'=1}^{C} \exp(m_{c'} + s_{c'}^2/2)}.$$

These can be used for gradient ascend to learn the parameters of $\tilde{q}$.

## A.7. Derivation for the Mixture Model

The marginal likelihood or evidence is

$$\exp \mathcal{L}_{\text{evd}} = \int p(\boldsymbol{u}) \sum_{\boldsymbol{c}} p(\boldsymbol{c}) \prod_{i=1}^{n} p(x_i|c_i, \boldsymbol{u}) \mathrm{d}\boldsymbol{u}.$$

Applying eq. (1) on $\mathcal{L}_{\text{evd}}$ focusing on the posterior for $p(\boldsymbol{u})$ gives

$$\mathcal{L}_{\text{evd}} \geq \frac{1}{1-\gamma} \log \int q(\boldsymbol{u}) \left[\sum_{\boldsymbol{c}} p(\boldsymbol{c}) \prod_{i=1}^{n} p(x_i|c_i, \boldsymbol{u})\right]^{1-\gamma} \mathrm{d}\boldsymbol{u} - \frac{\gamma}{1-\gamma} \log \int q(\boldsymbol{u})^{1/\gamma} p(\boldsymbol{u})^{1-1/\gamma} \mathrm{d}\boldsymbol{u}. \tag{6}$$

Applying ELBO on the logarithm of the term within the brackets above and then exponentiating the result get us to

$$\mathcal{L}_{\text{evd}} \geq \frac{1}{1-\gamma} \log \int q(\boldsymbol{u}) \left[\prod_{i=1}^{n} \prod_{c_i=1}^{K} \left(p(x_i|u_{c_i})\frac{p(c_i)}{q(c_i)}\right)^{q(c_i)}\right]^{1-\gamma} \mathrm{d}\boldsymbol{u} - \frac{\gamma}{1-\gamma} \log \int q(\boldsymbol{u})^{1/\gamma} p(\boldsymbol{u})^{1-1/\gamma} \mathrm{d}\boldsymbol{u}.$$

The argument of the logarithm in the first summand is the first displayed expression in section 3.1.3. In the first summand, we bring the products out of the logarithm:

$$\mathcal{L}_{\text{evd}} \geq \frac{1}{1-\gamma} \sum_{i=1}^{n} \sum_{c_i=1}^{K} \log \int q(\boldsymbol{u}) \left(p(x_i|u_{c_i})\frac{p(c_i)}{q(c_i)}\right)^{(1-\gamma)q(c_i)} \mathrm{d}\boldsymbol{u} - \frac{\gamma}{1-\gamma} \log \int q(\boldsymbol{u})^{1/\gamma} p(\boldsymbol{u})^{1-1/\gamma} \mathrm{d}\boldsymbol{u}.$$

In the first summand, the density ratios $p(c_i)/q(c_i)$ are independent of $\boldsymbol{u}$ and taken out to give the KL divergence. So the bound is written as

$$\frac{1}{1-\gamma} \sum_{i=1}^{n} \sum_{c_i=1}^{K} \log \int q(\boldsymbol{u}) \, p(x_i|u_{c_i})^{(1-\gamma)q(c_i)} \mathrm{d}\boldsymbol{u} - \sum_{i=1}^{n} \sum_{c_i=1}^{K} q(c_i) \log \frac{q(c_i)}{p(c_i)} - \frac{\gamma}{1-\gamma} \log \int q(\boldsymbol{u})^{1/\gamma} p(\boldsymbol{u})^{1-1/\gamma} \mathrm{d}\boldsymbol{u}.$$

Finally, use the mean-field approximation for $q(\boldsymbol{u})$ and rewriting the indexing for the $c_i$s as indexing for $k$ to give

$$\frac{1}{1-\gamma} \sum_{i=1}^{n} \sum_{k=1}^{K} \log \int q(u_k) \, p(x_i|u_k)^{(1-\gamma)q(c_i=k)} \mathrm{d}u_k - \sum_{i=1}^{n} \sum_{k=1}^{K} q(c_i = k) \log \frac{q(c_i = k)}{p(c_i = k)}$$

$$- \frac{\gamma}{1-\gamma} \sum_{k=1}^{K} \log \int q(u_k)^{1/\gamma} p(u_k)^{1-1/\gamma} \mathrm{d}u_k. \tag{7}$$

Swapping the order of the summations and using variational parameter $\varphi_{ik}$ for $q(c_i = k)$ gives the second displayed expression in section 3.1.3.

A.7.1. FRACTIONAL POSTERIORS FOR CLUSTER ASSIGNMENTS

For approximate inference to be tractable for the mixture model, it seems that we ultimately cannot avoid using ELBO for $\boldsymbol{c}$. Nonetheless, this does not preclude us from *also* having a fractional posterior for $\boldsymbol{c}$. Instead of applying ELBO on eq. (6), we apply the $\mathcal{L}^{\mathrm{b}}_{\gamma'}$ in section 2.3.2 to the same term:

$$\log \sum_{\boldsymbol{c}} p(\boldsymbol{c}) \prod_{i=1}^{n} p(x_i | c_i, \boldsymbol{u}) \geq \sum_{\boldsymbol{c}} r(\boldsymbol{c}) \log \prod_{i=1}^{n} p(x_i | c_i, \boldsymbol{u}) - \frac{1}{1 - \gamma'} \sum_{\boldsymbol{c}} r(\boldsymbol{c}) \log \frac{r(\boldsymbol{c})}{q(\boldsymbol{c})}$$
$$- \frac{\gamma'}{1 - \gamma'} \log \sum_{\boldsymbol{c}} q(\boldsymbol{c})^{1/\gamma'} p(\boldsymbol{c})^{1 - 1/\gamma'},$$

where $r(\boldsymbol{c})$ is the approximate Bayes posterior and $q(\boldsymbol{c})$ is the approximate fractional posterior. Following through derivations similar to before with mean-field approximations, we obtain

$$\mathcal{L}_{\mathrm{evd}} \geq \frac{1}{1 - \gamma} \sum_{i=1}^{n} \sum_{k=1}^{K} \log \int q(u_k) \, p(x_i | u_k)^{(1-\gamma) r(c_i=k)} \mathrm{d}u_k - \frac{1}{1 - \gamma'} \sum_{i=1}^{n} \sum_{k=1}^{K} r(c_i = k) \log \frac{r(c_i = k)}{q(c_i = k)}$$
$$- \frac{\gamma'}{1 - \gamma'} \sum_{i=1}^{n} \log \sum_{k=1}^{K} q(c_i = k)^{1/\gamma'} p(c_i = k)^{1 - 1/\gamma'} - \frac{\gamma}{1 - \gamma} \sum_{k=1}^{K} \log \int q(u_k)^{1/\gamma} p(u_k)^{1 - 1/\gamma} \mathrm{d}u_k.$$

As reasoned in section 2.3.2, the optimal fractional posteriors $q(c_i)$s interpolate between the $r(c_i)$s and the $p(c_i)$s: $q(c_i) \propto r(c_i)^{\gamma'} p(c_i)^{1-\gamma'}$. At this setting, we recover eq. (7) with a change of notation for the approximate Bayes posterior. This is expected since there is no constraint on fractional posteriors $q(c_i)$s other than normalisation.

# B. Monte Carlo Estimates

Suppose we have $N_{\mathrm{s}}$ samples $z_i$s from distribution $q(z) = \tilde{q}(z)/Z$ for known normalising constant $Z$. Then

$$\mathcal{L}_\gamma \approx \frac{1}{1 - \gamma} \log \frac{1}{N_{\mathrm{s}}} \sum_i p(D | z_i)^{1-\gamma} - \frac{\gamma}{1 - \gamma} \log \frac{1}{N_{\mathrm{s}}} \sum_i (q(z_i)/p(z_i))^{1/\gamma - 1}.$$

Ideally, one should draw separate samples for estimating $Z_{\mathrm{c}}$ and $Z_{\mathrm{d}}$, but in practice, one trades this off with computational efficiency. We currently employ this approach.

If we only have the unnormalised density, then approximating the lower bound requires the normalising factor $Z$:

$$\mathcal{L}_\gamma \approx \log Z + \frac{1}{1 - \gamma} \log \frac{1}{N_{\mathrm{s}}} \sum_i p(D | z_i)^{1-\gamma} - \frac{\gamma}{1 - \gamma} \log \frac{1}{N_{\mathrm{s}}} \sum_i \left( \frac{\tilde{q}(z_i)}{p(z_i)} \right)^{1/\gamma - 1}.$$

The normalising constant cannot be avoided, and one may estimate it with, for example, importance sampling from a distribution for which the normalising constant is known (Gelman & Meng, 1998).

An alternative is to introduce a mixing distribution and use the $\mathcal{L}^{\mathrm{h}}_\gamma$ bound:

$$\mathcal{L}^{\mathrm{h}}_\gamma \approx \frac{1}{1 - \gamma} \log \frac{1}{N_{\mathrm{s}}} \sum_i \sum_j p(D | z_{ij})^{1-\gamma} - \frac{\gamma}{1 - \gamma} \log \frac{1}{N_{\mathrm{s}}} \sum_i \sum_j \left( \frac{q(z_{ij} | u_i)}{p(z_{ij})} \right)^{1/\gamma - 1},$$

where there are now $N'_{\mathrm{s}}$ samples from $u_i \sim q(u)$ followed by $N_{\mathrm{s}}/N'_{\mathrm{s}}$ samples $z_{ij} \sim q(z | u_i)$. In practice, there can be different number of $z$ samples for each $u_i$, but we simplify the notation here. In this setting, $\tilde{q}(z)$ need not be known explicitly, but we are required to be able to draw the $(u_i, z_i)$s samples and to know the conditional $q(z | u)$ exactly.

For $\mathcal{L}^{\mathrm{bh}}_\gamma$, we similarly have $N'_{\mathrm{s}}$ samples from $u_i \sim q(u)$ and $N_{\mathrm{s}}/N'_{\mathrm{s}}$ samples $z_{ij} \sim q(z | u_i)$, but we now also have $N_{\mathrm{s}}/N'_{\mathrm{s}}$ samples $z'_{ik} \sim r(z | u_i)$:

$$\mathcal{L}^{\mathrm{bh}}_\gamma \approx \frac{1}{N_{\mathrm{s}}} \sum_i \sum_k \log p(D | z'_{ik}) - \frac{1}{1 - \gamma} \frac{1}{N_{\mathrm{s}}} \sum_i \sum_k \log \frac{r(z'_{ik} | u_i)}{q(z'_{ik} | u_i)} - \frac{\gamma}{1 - \gamma} \log \frac{1}{N_{\mathrm{s}}} \sum_i \sum_j \left( \frac{q(z_{ij} | u_i)}{p(z_{ij})} \right)^{1/\gamma - 1}.$$

For the alternative $\mathcal{L}_\gamma^{\text{bh}}$-alt bound (last row in Table 3), we suggest the following mechanism. We have the same samples, but further, for each $u_i$ sample, we associate with it a subset $U_i$ from the set $\{u_1, \ldots, u_{N'_{\text{s}}}\}$. Then we estimate this alternate bound with

$$\frac{1}{N_{\text{s}}} \sum_i \sum_k \log p(D|z'_{ik}) - \frac{1}{1-\gamma} \frac{1}{N_{\text{s}}} \sum_i \sum_k \log r(z'_{ik}|u_i)$$

$$+ \frac{1}{1-\gamma} \frac{1}{N_{\text{s}}} \sum_i \sum_k \frac{1}{|U_i|} \sum_{u_j \in U_i} \log q(z'_{ik}|u_j) - \frac{\gamma}{1-\gamma} \log \frac{1}{N_{\text{s}}} \sum_i \sum_j \left( \frac{q(z_{ij}|u_i)}{p(z_{ij})} \right)^{1/\gamma - 1}.$$

If $q(u)$ is a continuous distribution such that there is zero probability of having two samples with the same value, then it is important that $U_i$ excludes $u_i$. Similarly, if $|U_i|$ is small, then a systematic instead of random selection of the subsamples should be better. The method of unbiased implicit variational inference (Titsias & Ruiz, 2019) similarly requires a nested summation of independent samples from $q(u)$, but the proposal there is to sample $U_i$ separately. The trade-off between computational cost and more faithful Monte Carlo estimates depends on, for example, the cost of sampling from $q(u)$. Yet another approach is the Gaussian approximation based on linearisation (Uppal et al., 2023).

### B.1. On Single Samples

If $N_{\text{s}} = 1$, as commonly done for local latent variable models (see section 5.2), then the above estimates for $\mathcal{L}_\gamma$ and $\mathcal{L}_\gamma^{\text{h}}$ revert to ELBO because the exponents within the logarithms cancel the multiplicative factors on the logarithms. Therefore, these bounds require $N_{\text{s}} > 1$ to be effective.

For $\mathcal{L}_\gamma^{\text{bh}}$, the $1/(1-\gamma)$ factor remains for the KL divergence from $r$ to $q$. The implications may be subject to future work.

### B.2. Considerations for Learning

If we are using the above estimates within an automatic differentiation procedure for learning the parameters of the distributions, it is critical that any sampling distributions (that is, $q(z)$, or $q(z|u)$ and $q(u)$) be driven from standard distributions with fixed parameters so that we may apply the *Law of the Unconscious Statistician*, also known as the reparameterisation trick (Kingma & Welling, 2014).

In addition, for the semi-implicit constructions, we find learning to be effective when $N_{\text{s}}/N'_{\text{s}} > 1$, that is more than one sample of $z$ for each sample of $u$.

For ELBO with semi-implicit posteriors, although a delta distribution is known to be optimal (Yin & Zhou, 2018), variances in sampling may lead to a mixture of delta distributions located at the optimal and the near-optimal locations. This is further exacerbated by the variance in stochastic gradient methods. The parameterisation of the implicit posteriors may also give a region of lower probabilities "bridging" these locations. Fig. 6 in section C.4 provides an illustration based on VAE on the MNIST data set.

## C. Experiments

This section collects additional details and results for the experiments.

### C.1. Calibration

For the calibration experiment (section 5.1), we initialise the estimated posterior with means $-1$ and $1$, and variances $n/K$. The prior standard deviations of the component means are 3. The experiment is not sensitive to the precise settings of these quantities.

#### C.1.1. STRATEGY USING INVERSE SQUARED LENGTH

For a prior $p(z)$ and a corresponding exact Bayes posterior $p(z|D)$, the exact fractional posterior is given by the interpolation $p(z)^{1-\gamma} p(z|D)^\gamma$. For Gaussian distributions, the precision of this fractional posterior is $(1-\gamma)/\sigma_0^2 + \gamma/\sigma_1^2$, where $\sigma_0^2$ and $\sigma_1^2$ are the variances of the prior and the Bayes posterior. In the context of calibration, precision is inversely proportional to the square of the interval length.

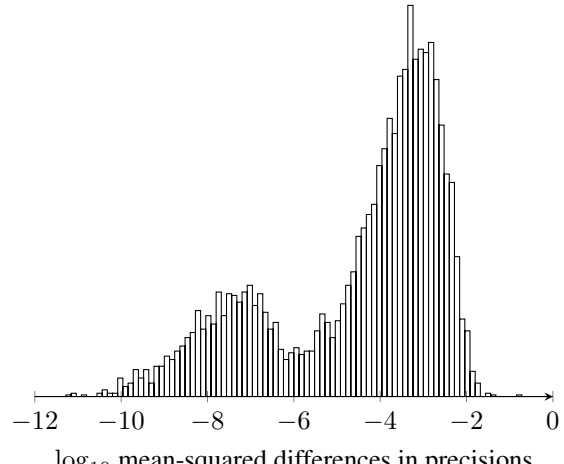

(a) The histogram of the logarithm of the mean-squared differences.

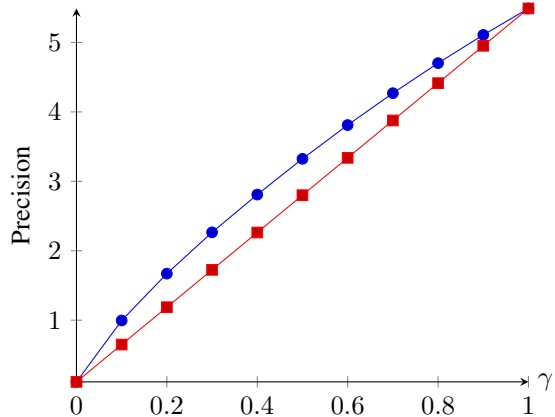

(b) An example of precisions for one data set. The straight line (with squares) is the interpolated precision, while the curved line (with circles) is obtained from the posterior learnt with $\mathcal{L}_\gamma$.

Figure 2: Differences between the precisions of the fractional posteriors for the first component of the mixture model.

This suggests us to perform linear regression against $\ell^{-2}$ to target the required interval length, using the results from the approximate posteriors. For the experimental setup in section 5.1, this strategy, called $R_{\ell^{-2}}$, gives $\kappa$s (resp. $\ell$s) with $0.9320$ (resp. $0.2735$) and $0.9220$ (resp. $0.2888$) for $\mu_1$ and $\mu_2$. While $R_{\ell^{-2}}$ does gives interval lengths closest to the ideal of $0.2772$, the coverages are lower. The $\gamma$ for this is $0.974$ with evidence bound $-834.9$.

That strategy $R_{\ell^{-2}}$ is not necessary best in all respects reminds us that we are dealing with a mixture model and not a Gaussian model. The difference is empirically elaborated in section C.2.

### C.1.2. DIFFERENT EXPERIMENTAL SETTINGS

We find the results and conclusions to be similar when the number of observations $n$ per data set is different, albeit with different interval lengths. Table 4a gives the results for the same experiment but with $n = 30$ (versus the 400 in section 5.1). This is similar for closer component means at $-1/2$ and $1/2$, though now with different coverages (Table 4b), and only the more comprehensive $R_\kappa$ that gives $\gamma = 0.38$ is effective for calibration in this more difficult setting.

The picture is more complex with $K = 4$ components, centred at $-2$, $-1/2$, $1/2$ and $2$:[1] the coverages for components at $\pm 1/2$ are noticeably smaller than for components at $\pm 2$, more so for larger $\gamma$s (Table 4c). Again, only $R_\kappa$ that gives $\gamma = 0.26$ is close to effective for calibration.

This extended study highlights the need to consider fractional posteriors, especially for difficult problems.

### C.2. Differences with Fractional Posteriors obtained by Interpolation

We seek to illustrate that a fractional posterior obtained by our bounds is in generally different from that obtained by interpolating between the prior and the Bayes posterior (see section C.1.1). We follow section 5.1 and section C.1, but with $K = 2$, $n = 20$ and component centres $-1/2$ and $1/2$ in order to make the differences more noticeable.

With compute the approximate Bayes posterior $r$ learnt by optimising ELBO ($\mathcal{L}_{1.0}$), we interpolate with prior to obtain a fractional posteriors $p^{1-\gamma}r^\gamma$. This is compared with the approximate fractional posterior $q_\gamma$ obtained from optimising $\mathcal{L}_\gamma$. It is sufficient to examine the precisions for the first component to show the differences.

We compute mean-squared differences between the precisions of the two set of fractional posteriors at $\gamma \in \{0.1, 0.2, 0.3, 0.4, 0.5, 0.6, 0.7, 0.8, 0.9\}$. Figure 2a plots the histogram, across the 5000 data sets, of the base-10 logarithm of the mean-squared differences; and Fig. 2b plots the precisions for a particular data set. The plots demonstrate that the interpolated posteriors and the learnt posteriors are different, in general.

---

[1]The posterior means are initialised at $\pm 1$ and $\pm 1/4$.

Table 4: Calibration study of the Gaussian mixture model at $\alpha = 0.05$ significance level, for different settings.

(a) For $n = 30$. The $\gamma$ values for $R_\ell$, $R_{\ell-2}$ and $R_\kappa$ are 0.789, 0.965 and 0.849.

| | $\mu_1$ | | $\mu_2$ | | |
| --- | --- | --- | --- | --- | --- |
| | $\kappa$ | $\ell$ | $\kappa$ | $\ell$ | bound |
| $\mathcal{L}_{0.1}$ | 1.0000 | 3.0111 | 1.0000 | 3.1774 | $-69.2$ |
| $\mathcal{L}_{0.2}$ | 0.9998 | 2.1644 | 0.9998 | 2.2901 | $-69.1$ |
| $\mathcal{L}_{0.3}$ | 0.9990 | 1.7770 | 0.9978 | 1.8822 | $-69.4$ |
| $\mathcal{L}_{0.4}$ | 0.9968 | 1.5433 | 0.9936 | 1.6355 | $-69.7$ |
| $\mathcal{L}_{0.5}$ | 0.9922 | 1.3826 | 0.9876 | 1.4658 | $-69.9$ |
| $\mathcal{L}_{0.6}$ | 0.9844 | 1.2636 | 0.9780 | 1.3399 | $-70.2$ |
| $\mathcal{L}_{0.7}$ | 0.9746 | 1.1708 | 0.9672 | 1.2418 | $-70.4$ |
| $\mathcal{L}_{0.8}$ | 0.9620 | 1.0958 | 0.9524 | 1.1624 | $-70.5$ |
| $\mathcal{L}_{0.9}$ | 0.9498 | 1.0336 | 0.9382 | 1.0966 | $-70.7$ |
| $\mathcal{L}_{1.0}$ | 0.9328 | 0.9810 | 0.9234 | 1.0408 | $-70.8$ |
| $C_{0.1}$ | 1.0000 | 3.0111 | 1.0000 | 3.1774 | $-74.7$ |
| $C_{0.5}$ | 0.9920 | 1.3826 | 0.9874 | 1.4658 | $-70.4$ |
| $C_{0.9}$ | 0.9494 | 1.0336 | 0.9378 | 1.0966 | $-70.7$ |
| $R_\ell$ | 0.9628 | 1.1035 | 0.9538 | 1.1706 | $-70.5$ |
| $R_{\ell-2}$ | 0.9398 | 0.9983 | 0.9296 | 1.0595 | $-70.8$ |
| $R_\kappa$ | 0.9570 | 1.0642 | 0.9464 | 1.1290 | $-70.6$ |

(b) For $\mu_1 = -1/2$, $\mu_2 = 1/2$. The $\gamma$ values for $R_\ell$, $R_{\ell-2}$ and $R_\kappa$ are 0.785, 0.998 and 0.379.

| | $\mu_1$ | | $\mu_2$ | | |
| --- | --- | --- | --- | --- | --- |
| | $\kappa$ | $\ell$ | $\kappa$ | $\ell$ | bound |
| $\mathcal{L}_{0.1}$ | 0.9996 | 0.8740 | 0.9986 | 0.8743 | $-625.8$ |
| $\mathcal{L}_{0.2}$ | 0.9946 | 0.6188 | 0.9898 | 0.6190 | $-624.9$ |
| $\mathcal{L}_{0.3}$ | 0.9828 | 0.5055 | 0.9682 | 0.5057 | $-624.9$ |
| $\mathcal{L}_{0.4}$ | 0.9642 | 0.4379 | 0.9434 | 0.4380 | $-625.0$ |
| $\mathcal{L}_{0.5}$ | 0.9382 | 0.3917 | 0.9176 | 0.3918 | $-625.2$ |
| $\mathcal{L}_{0.6}$ | 0.9082 | 0.3576 | 0.8884 | 0.3577 | $-625.3$ |
| $\mathcal{L}_{0.7}$ | 0.8816 | 0.3311 | 0.8598 | 0.3312 | $-625.4$ |
| $\mathcal{L}_{0.8}$ | 0.8558 | 0.3097 | 0.8378 | 0.3098 | $-625.6$ |
| $\mathcal{L}_{0.9}$ | 0.8320 | 0.2920 | 0.8212 | 0.2921 | $-625.7$ |
| $\mathcal{L}_{1.0}$ | 0.8126 | 0.2771 | 0.7972 | 0.2771 | $-625.8$ |
| $C_{0.1}$ | 0.9990 | 0.8740 | 0.9978 | 0.8743 | $-630.2$ |
| $C_{0.5}$ | 0.9372 | 0.3917 | 0.9176 | 0.3918 | $-625.4$ |
| $C_{0.9}$ | 0.8314 | 0.2920 | 0.8212 | 0.2921 | $-625.7$ |
| $R_\ell$ | 0.8596 | 0.3127 | 0.8414 | 0.3128 | $-625.6$ |
| $R_{\ell-2}$ | 0.8128 | 0.2773 | 0.7974 | 0.2774 | $-625.8$ |
| $R_\kappa$ | 0.9686 | 0.4501 | 0.9486 | 0.4503 | $-625.0$ |

(c) For $K = 4$, with component means $-2, -1/2, 1/2, 2$. The $\gamma$ values for $R_\ell$, $R_{\ell-2}$ and $R_\kappa$ are 0.785, 0.995 and 0.263.

| | $\mu_1$ | | $\mu_2$ | | $\mu_3$ | | $\mu_4$ | | |
| --- | --- | --- | --- | --- | --- | --- | --- | --- | --- |
| | $\kappa$ | $\ell$ | $\kappa$ | $\ell$ | $\kappa$ | $\ell$ | $\kappa$ | $\ell$ | bound |
| $\mathcal{L}_{0.1}$ | 0.9990 | 1.2259 | 0.9876 | 1.2369 | 0.9696 | 1.2377 | 0.9994 | 1.2309 | $-820.7$ |
| $\mathcal{L}_{0.2}$ | 0.9942 | 0.8712 | 0.9046 | 0.8757 | 0.8904 | 0.8757 | 0.9956 | 0.8740 | $-817.3$ |
| $\mathcal{L}_{0.3}$ | 0.9862 | 0.7126 | 0.7980 | 0.7152 | 0.8310 | 0.7151 | 0.9814 | 0.7147 | $-816.7$ |
| $\mathcal{L}_{0.4}$ | 0.9710 | 0.6176 | 0.7188 | 0.6195 | 0.7864 | 0.6194 | 0.9660 | 0.6194 | $-816.7$ |
| $\mathcal{L}_{0.5}$ | 0.9514 | 0.5527 | 0.6558 | 0.5542 | 0.7410 | 0.5540 | 0.9434 | 0.5542 | $-816.8$ |
| $\mathcal{L}_{0.6}$ | 0.9328 | 0.5047 | 0.6076 | 0.5059 | 0.7018 | 0.5057 | 0.9234 | 0.5061 | $-817.0$ |
| $\mathcal{L}_{0.7}$ | 0.9120 | 0.4674 | 0.5746 | 0.4684 | 0.6656 | 0.4682 | 0.9020 | 0.4687 | $-817.2$ |
| $\mathcal{L}_{0.8}$ | 0.8878 | 0.4373 | 0.5436 | 0.4382 | 0.6384 | 0.4380 | 0.8846 | 0.4385 | $-817.4$ |
| $\mathcal{L}_{0.9}$ | 0.8654 | 0.4123 | 0.5126 | 0.4131 | 0.6126 | 0.4130 | 0.8660 | 0.4134 | $-817.6$ |
| $\mathcal{L}_{1.0}$ | 0.8422 | 0.3912 | 0.4864 | 0.3919 | 0.5970 | 0.3918 | 0.8484 | 0.3923 | $-817.8$ |
| $C_{0.1}$ | 0.9988 | 1.2259 | 0.9830 | 1.2369 | 0.9684 | 1.2377 | 0.9996 | 1.2309 | $-826.4$ |
| $C_{0.5}$ | 0.9504 | 0.5527 | 0.6530 | 0.5542 | 0.7412 | 0.5540 | 0.9460 | 0.5542 | $-817.0$ |
| $C_{0.9}$ | 0.8650 | 0.4123 | 0.5106 | 0.4131 | 0.6174 | 0.4130 | 0.8664 | 0.4134 | $-817.6$ |
| $R_\ell$ | 0.8922 | 0.4414 | 0.5478 | 0.4423 | 0.6412 | 0.4421 | 0.8860 | 0.4426 | $-817.4$ |
| $R_{\ell-2}$ | 0.8432 | 0.3922 | 0.4884 | 0.3929 | 0.5926 | 0.3927 | 0.8500 | 0.3932 | $-817.8$ |
| $R_\kappa$ | 0.9904 | 0.7610 | 0.8404 | 0.7642 | 0.8506 | 0.7641 | 0.9870 | 0.7633 | $-816.8$ |

Table 5: Importance sampling to estimate the log-evidence $\widehat{\mathcal{L}}_{\mathrm{evd}}$ of Gaussian mixture models using approximate posteriors. We use the four experimental settings from section 5.1 and section C.1.2. For each setting, and for ten values of $\gamma$ (including $\gamma = 1.0$ for ELBO), we have the bound $\mathcal{L}_\gamma$ from the variational optimisation, $\widehat{\mathcal{L}}_{\mathrm{evd}}$, and the coefficient of variation CV of the weights used to compute $\widehat{\mathcal{L}}_{\mathrm{evd}}$.

| $\gamma$ | $n = 400, \mu_i = \pm 2$ | | | $n = 30, \mu_i = \pm 2$ | | | $n = 400, \mu_i = \pm 1/2$ | | | $n = 400$, 4 components | | |
|---|---|---|---|---|---|---|---|---|---|---|---|---|
| | $\mathcal{L}_\gamma$ | $\widehat{\mathcal{L}}_{\mathrm{evd}}$ | CV | $\mathcal{L}_\gamma$ | $\widehat{\mathcal{L}}_{\mathrm{evd}}$ | CV | $\mathcal{L}_\gamma$ | $\widehat{\mathcal{L}}_{\mathrm{evd}}$ | CV | $\mathcal{L}_\gamma$ | $\mathcal{L}_{\mathrm{evd}}$ | CV |
| 0.1 | $-813.0$ | $-806.9$ | 2.2 | $-65.7$ | $-60.2$ | 2.0 | $-602.1$ | $-593.1$ | 84.0 | $-805.2$ | $-786.3$ | 87.6 |
| 0.2 | $-812.9$ | $-806.9$ | 1.5 | $-64.6$ | $-60.2$ | 1.3 | $-601.0$ | $-593.4$ | 23.7 | $-801.8$ | $-786.3$ | 83.0 |
| 0.3 | $-813.2$ | $-806.9$ | 1.1 | $-64.5$ | $-60.2$ | 1.0 | $-600.9$ | $-593.4$ | 19.8 | $-801.1$ | $-786.2$ | 62.8 |
| 0.4 | $-813.5$ | $-806.9$ | 1.0 | $-64.5$ | $-60.2$ | 0.8 | $-601.0$ | $-593.4$ | 31.5 | $-801.1$ | $-786.4$ | 36.8 |
| 0.5 | $-813.7$ | $-806.9$ | 0.9 | $-64.6$ | $-60.2$ | 0.6 | $-601.2$ | $-593.4$ | 24.3 | $-801.2$ | $-786.6$ | 19.4 |
| 0.6 | $-813.9$ | $-806.9$ | 0.9 | $-64.7$ | $-60.2$ | 0.5 | $-601.3$ | $-592.8$ | 163.0 | $-801.4$ | $-786.2$ | 83.9 |
| 0.7 | $-814.1$ | $-806.9$ | 1.0 | $-64.8$ | $-60.2$ | 0.4 | $-601.4$ | $-592.8$ | 130.9 | $-801.6$ | $-786.6$ | 26.6 |
| 0.8 | $-814.3$ | $-806.9$ | 0.8 | $-64.9$ | $-60.2$ | 0.3 | $-601.6$ | $-593.5$ | 12.0 | $-801.8$ | $-786.8$ | 16.9 |
| 0.9 | $-814.4$ | $-806.9$ | 1.1 | $-65.0$ | $-60.2$ | 0.3 | $-601.7$ | $-593.2$ | 96.9 | $-802.0$ | $-786.8$ | 17.0 |
| 1.0 | $-814.5$ | $-806.9$ | 1.2 | $-65.1$ | $-60.2$ | 0.3 | $-601.8$ | $-593.6$ | 15.5 | $-802.2$ | $-786.8$ | 23.0 |

## C.3. The Tightness of Bounds for the Gaussian Mixture Models

We use importance sampling to estimate the evidence of the Gaussian mixture models (GMMs) using approximate posteriors. For the log-evidence of $-827.24$ reported in section 5.1, we draw $N_{\mathrm{s}} = 1,000$ samples from the approximate posteriors $q(\boldsymbol{u}, \boldsymbol{c})$ to estimate the log-evidence $\widehat{\mathcal{L}}_{\mathrm{evd}} \stackrel{\mathrm{def}}{=} \log \sum_{i=1}^{N_{\mathrm{s}}} w_i / N_{\mathrm{s}}$, where weights $w_i \stackrel{\mathrm{def}}{=} p(\boldsymbol{u}_i, \boldsymbol{c}_i) p(\boldsymbol{x}|\boldsymbol{u_i}, \boldsymbol{c}_i) / q(\boldsymbol{u}_i, \boldsymbol{c}_i)$. The average log-evidence of $-827.24$ is obtained over the $5,000$ replications. The same value, up to five significant figures, is obtain with every one of the ten fractional or Bayes posteriors.

We perform more experiments with four settings: the one in section 5.1 and the three in section C.1.2. We use one of the $5,000$ replications for this. For each setting, we draw $N_{\mathrm{s}} = 100,000$ importance samples from the approximate posteriors $q(\boldsymbol{u}, \boldsymbol{c})$ to estimate $\widehat{\mathcal{L}}_{\mathrm{evd}}$. We also compute the coefficient of variation (CV) of the weights to give an indication of the quality of the estimates. By central limit theorem, this is $\sqrt{N_{\mathrm{s}}}$ times the the coefficient of variation of the evidence $\exp \widehat{\mathcal{L}}_{\mathrm{evd}}$.

The results demonstrate that there is no guarantee that lower $\gamma$ will give better bounds (columns $\mathcal{L}_\gamma$ in Table 5), as we have reasoned in section 2.2, though the best bounds are typically not with ELBO ($\gamma = 1.0$).

There is also a substantial gap between $\widehat{\mathcal{L}}_{\mathrm{evd}}$ and the best bounds (comparing columns $\mathcal{L}_\gamma$ and $\widehat{\mathcal{L}}_{\mathrm{evd}}$ in Table 5). We opine that this is because (1) the approximate posterior $q(\boldsymbol{u}, \boldsymbol{c})$ is factorised into $q(\boldsymbol{u})q(\boldsymbol{c})$ where only $q(\boldsymbol{u})$ is the approximate fractional posterior while $q(\boldsymbol{c})$ is still the approximate Bayes posterior; and (2) in our GMM settings $\boldsymbol{c}$ has a larger role because there are $n \geq 30$ data points while for $\boldsymbol{u}$ we have at most 4 one-dimensional components.

For the quality of importance sampling, the CVs are significantly smaller when the true means are more separated (first two columns of CV versus the last two columns of CV in Table 5). This is expected because better separation suggests multi-modality in the true Bayes posterior is less pronounced. In addition, for the purpose of better importance sampling in GMMs to obtain $\widehat{\mathcal{L}}_{\mathrm{evd}}$, using $\mathcal{L}_\gamma$ has only a slight advantage — we attribute this again to $q(\boldsymbol{c})$ being an approximate Bayes posterior and that $\boldsymbol{c}$ is a larger role.

If the end goal is to perform better importance sampling using fractional posteriors, we may use the fractional posteriors for the cluster assignments, which are interpolations between the approximate Bayes posteriors and the priors (section A.7.1).

## C.4. Variational Autoencoder

We make three experimental changes from Ruthotto & Haber (2021). One, we use the *continuous Bernoulli distribution* (Loaiza-Ganem & Cunningham, 2019) as the likelihood function instead of the cross-entropy loss function. Two, we draw 100 samples from the approximate posterior per datum during training instead of the one sample that they use, because otherwise there is no difference between the ELBO and some of our bounds (see section B.1). Three, we train for 500

Table 6: Average log-evidences (higher better) over data samples, and its breakdown for VAE on MNIST data sets, for $\mathcal{L}_\gamma^{\text{bh}}$ and $\mathcal{L}_\gamma^{\text{bh}}$-alt. We give the mean and *three* standard deviations of these averages over ten experimental runs. For Monte Carlo averages, 1,024 samples are used ($32 \times 32$ for semi-implicit posteriors). We show the addition of the KL divergence and Rényi's divergence for *Test using Objective*; and for *Test using ELBO* we evaluate the Bayes posterior $r$.

| Objective | $\gamma$ | Train (Total) | Test using Objective | | | Test using ELBO | | |
|---|---|---|---|---|---|---|---|---|
| | | | Total | data | div | Total | data | div |
| $\mathcal{L}_\gamma^{\text{bh}}$ | 0.9 | $1636.2_{\pm 11.5}$ | $1608.8_{\pm 23.3}$ | $1613.4_{\pm 23.0}$ | $4.0_{\pm 0.2} + 0.6_{\pm 0.3}$ | $1607.7_{\pm 23.9}$ | $1613.4_{\pm 23.0}$ | $5.7_{\pm 1.0}$ |
| | 0.5 | $1635.2_{\pm 10.5}$ | $1608.0_{\pm 25.4}$ | $1612.7_{\pm 25.2}$ | $4.3_{\pm 0.2} + 0.4_{\pm 0.2}$ | $1607.4_{\pm 25.8}$ | $1612.7_{\pm 25.3}$ | $5.3_{\pm 0.6}$ |
| | 0.1 | $1635.5_{\pm 12.0}$ | $1607.5_{\pm 16.1}$ | $1612.4_{\pm 16.1}$ | $4.6_{\pm 0.1} + 0.3_{\pm 0.2}$ | $1607.3_{\pm 16.2}$ | $1612.4_{\pm 16.1}$ | $5.1_{\pm 0.2}$ |
| $\mathcal{L}_\gamma^{\text{bh}}$-alt | 0.9 | $1639.4_{\pm 9.4}$ | $1609.1_{\pm 21.1}$ | $1613.6_{\pm 21.0}$ | $4.0_{\pm 0.1} + 0.6_{\pm 0.2}$ | $1608.0_{\pm 21.6}$ | $1613.6_{\pm 21.0}$ | $5.6_{\pm 0.8}$ |
| | 0.5 | $1639.4_{\pm 10.6}$ | $1608.2_{\pm 24.6}$ | $1612.9_{\pm 24.5}$ | $4.3_{\pm 0.1} + 0.4_{\pm 0.2}$ | $1607.6_{\pm 24.8}$ | $1612.9_{\pm 24.5}$ | $5.3_{\pm 0.5}$ |
| | 0.1 | $1639.2_{\pm 10.4}$ | $1608.8_{\pm 26.6}$ | $1613.7_{\pm 26.3}$ | $4.7_{\pm 0.1} + 0.3_{\pm 0.2}$ | $1608.6_{\pm 26.6}$ | $1613.7_{\pm 26.3}$ | $5.1_{\pm 0.3}$ |

epochs instead of the 50 epochs that they have used, so that we obtain results closer to convergence to reduce doubts on the comparisons.

For the semi-implicit posterior family, we use three layers neural network for the implicit distribution, similar to that used by Yin & Zhou (2018) with the following changes: we reduce the noise dimensions to 15, 10 and 5, and the hidden dimensions to 28, 14 and 2 so that we can visualise the samples from the implicit distribution; we use normal with mean 0.5 and standard deviation 1 instead of Bernoulli for the noise distribution to better match the gray-scale images that we use; we use leaky ReLU activations (Maas et al., 2013) for the hidden units to reduce degeneracy due to the learning dynamics, and we use sigmoid for the output unit so that we can visualise the distribution within a unit square.

For training the explicit posteriors with $\mathcal{L}_\gamma$, we use 100 samples per datum. For the semi-implicit posteriors with $\mathcal{L}_\gamma^{\text{h}}$, we draw 10 samples from the implicit distribution per datum, then 10 latent variables from the explicit distribution per implicit sample. For $\mathcal{L}_\gamma^{\text{bh}}$ involving semi-implicit fractional and Bayes posteriors, we additionally use 5 of the 10 implicit samples to estimate the cross entropy term. Batch size of 64 is used for training with the Adam optimizer, where the learning rate and weight decay are set to $10^{-3}$ and $10^{-5}$.

We use ten experimental runs to obtain Table 2. The neural networks in each run are iniitlised with a different seed for the pseudo-random number generator. The results in the table are the mean and three standard deviations of these ten runs. The overall relatively small variations among the ten runs suggests the stability of the results.

We also perform the same experiments for the $\mathcal{L}_\gamma^{\text{bh}}$-alt bound (last row of Table 3), and the results are compared with those of $\mathcal{L}_\gamma^{\text{bh}}$ in Table 6). We find their results to be very similar.

**Bounds** We take a single run of $\mathcal{L}_{\text{ELBO}}$ for the explicit posterior and uses its decoder as the fixed decoder to train the encoders (or posteriors) for $\mathcal{L}_\gamma$ for $\gamma \in \{0.1, 0.5, 0.9, 1.0\}$. The encoder neural networks are initialised randomly and optimised for 50 epochs with the same number of Monte Carlo as before during training. Since the decoder and hence likelihood models are now fixed together with the prior, we may compare the train and test objectives as bounds on a single fixed log-evidence. With smaller $\gamma$, we obtain tighter evidence bounds and posteriors closer to the prior (second, third and eighth columns of Table 7). The results for the reference $\mathcal{L}_{\text{ELBO}}$ (that is, first row in the table with (reference) 1.0) are that for joint optimisation with the decoder for 500 epochs, and are used for Table 2. Comparing the figures for the two instances of $\gamma = 1.0$, we see that the dynamics of jointly training decoder and encoder can give better objectives.

**Image generation** We perform further investigation by plotting figures from a single run each. Figure 3 gives the images from the learnt decoders for the VAE experiments using latent variables sampled from the *prior* in a regular manner. We do not see any significant quality to the decoded images from the different values of $\gamma$ tried. Differences are more visible for the harder Fashion-MINST data set (section 5.3). There are two reasons why the images in the figure (except for Fig. 3i) are not sharper: (1) we use simple neural networks for the decoder and encoders (Ruthotto & Haber, 2021) with only 88,837 parameters for the case of $\mathcal{L}_\gamma$; and (2) the images are *mean* images from the decoded latent variables, that is, the parameters of the continuous Bernoulli distributions, and not samples.

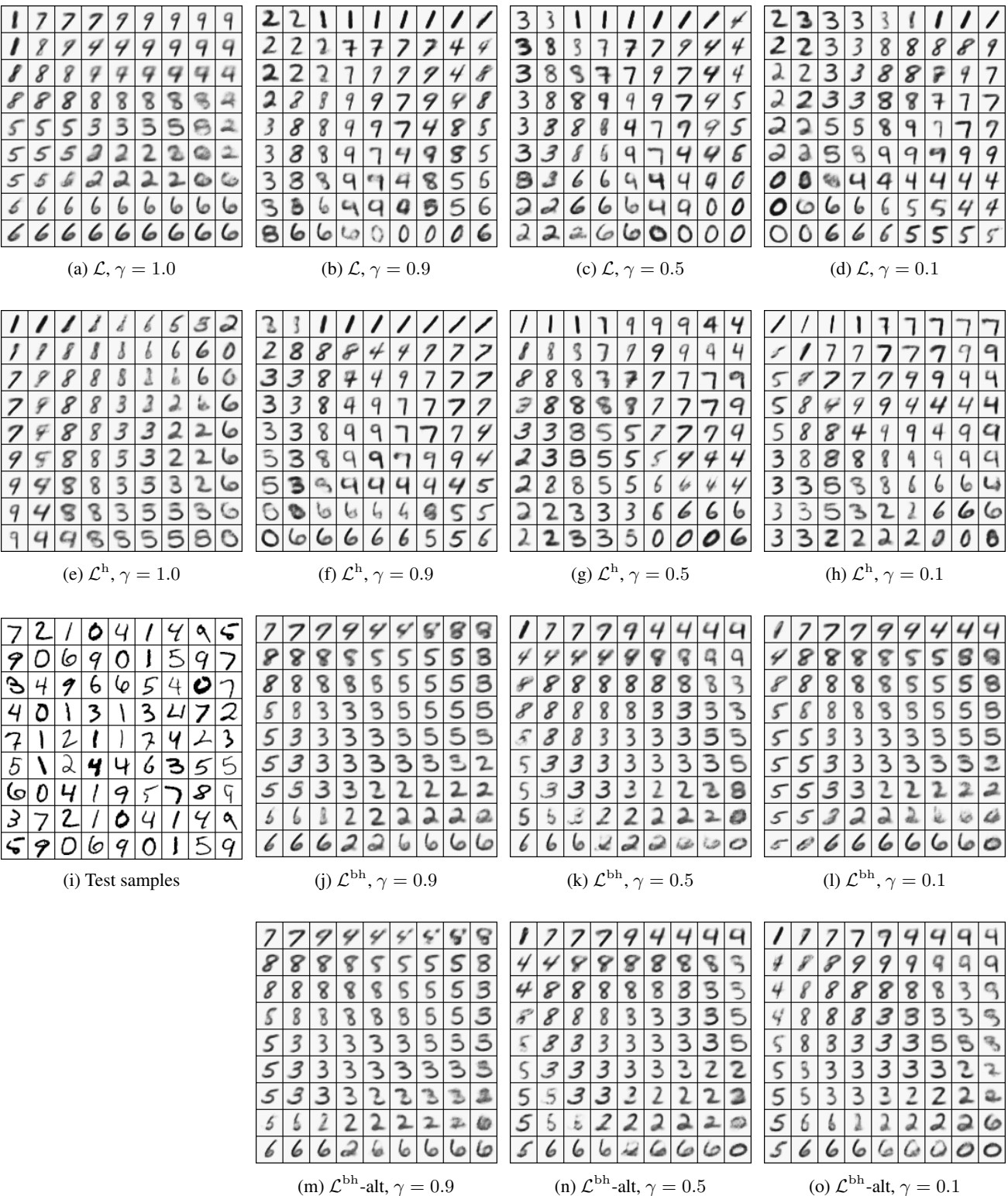

Figure 3: Mean images from decoded latent variables obtained by coordinate-wise inverse-CDF (standard normal) transform from a unit square. For the last row, the first image are samples from the test set. The quality of the images are visually similar across $\gamma$s and posterior families, and they are all not as sharp as the real test images. The VAE is trained on the MNIST dataset.

Table 7: Log-evidences (higher better) over data samples for a single run, and its breakdown for VAE on MNIST data sets, for $\mathcal{L}_\gamma$ where the decoders are fixed to be the same as that optimised for ELBO (first row in table). For Monte Carlo averages, 1,024 samples are used.

| $\gamma$ | Train (Total) | Test using Objective | | | Test using ELBO | | |
|---|---|---|---|---|---|---|---|
| | | Total | data | div | Total | data | div |
| (reference) 1.0 | 1616.814 | 1591.401 | 1596.695 | 5.294 | 1591.407 | 1596.700 | 5.293 |
| 1.0 | 1580.006 | 1558.188 | 1562.939 | 4.751 | 1558.189 | 1562.940 | 4.751 |
| 0.9 | 1625.566 | 1620.629 | 1624.170 | 3.541 | 1551.088 | 1554.442 | 3.354 |
| 0.5 | 1638.652 | 1636.510 | 1639.442 | 2.933 | 1451.295 | 1453.317 | 2.022 |
| 0.1 | 1641.655 | 1639.472 | 1642.944 | 3.472 | 1427.995 | 1429.871 | 1.876 |

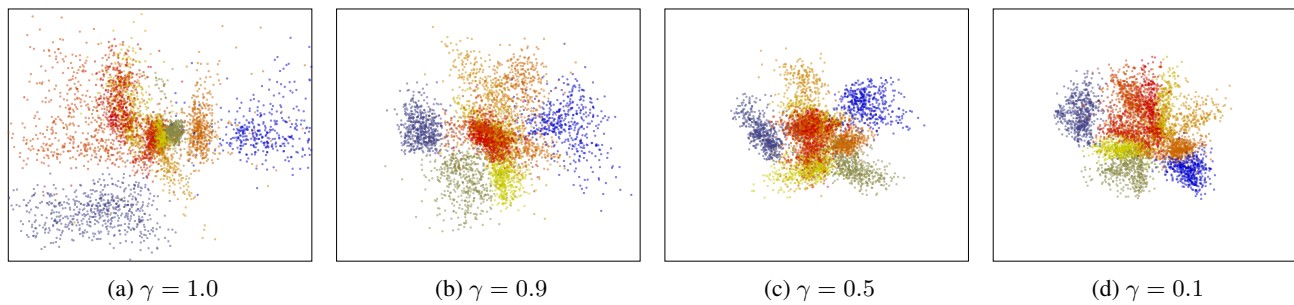

(a) $\gamma = 1.0$         (b) $\gamma = 0.9$         (c) $\gamma = 0.5$         (d) $\gamma = 0.1$

Figure 4: Means of the explicit posteriors for 5,000 sampled MNIST test images, colour-coded by the class labels. All axes ranges from $-4$ to $4$.

**Posteriors**   Figure 4 gives the means of the explicit posterior, and Fig. 5 gives the samples from the posteriors, explicit and semi-implicit. For $\mathcal{L}_\gamma$ and $\mathcal{L}_\gamma^{\mathrm{h}}$, the posteriors are visually closer to the prior for smaller gamma, except for between $\gamma = 0.5$ and 0.1: their KL divergences are similar in Table 2. The scatter plots for $\mathcal{L}_\gamma^{\mathrm{bh}}$ and $\mathcal{L}_\gamma^{\mathrm{bh}}$-alt (Figs. 5i to 5n) are for the Bayes posteriors, and they seems to have more clumps than $\mathcal{L}_{1.0}^{\mathrm{h}}$'s (Fig. 5e).

Figure 6 plots the samples from the implicit distributions of $\mathcal{L}_\gamma^{\mathrm{h}}$, $\mathcal{L}_\gamma^{\mathrm{bh}}$ and $\mathcal{L}_\gamma^{\mathrm{bh}}$-alt separately for 16 test images, that is, one plot for each test image in each setting. We see diversity in the implicit distributions in majority of the cases for $\mathcal{L}_\gamma^{\mathrm{h}}$ with $\gamma < 1.0$ (Figs. 6b to 6d). This is diversity is seldom for ELBO ($\mathcal{L}_{1.0}^{\mathrm{h}}$, Fig. 6a), but not totally absent despite the theory for otherwise (Yin & Zhou, 2018), probably because of noise in learning with Monte Carlo samples and the neural network parameters giving similar optimum values.

For $\mathcal{L}_\gamma^{\mathrm{bh}}$ and $\mathcal{L}_\gamma^{\mathrm{bh}}$-alt (Figs. 6e to 6j), we see almost lack of diversity in the implicit samples. We attribute this to the larger magnitude of the data-fit term in the objective over the divergence (about 340 times larger in Table 2) causing the theoretical degeneracy of the ELBO to be prominent.

To have a broad overview of the distributions of the implicits, we compute the sample covariance for the implicit distributions of each test image with 500 samples after transforming with arcsine-square-root. Each sample covariance is used to compute the generalised variance and total variation, and we summarise them over the 10,000 test images using the following descriptive statistics (Table 8): median, maximum, mean, coefficient of variation (CV) and skewness (Fisher-Pearson coefficient). We caution that this assumes single modes in the two-dimensional implicit distributions. We find the medians, maximums and means of the generalised variances to be significantly smaller than that of the total variations, which indicates that most distributions approximately degenerate and can hardly be considered two-dimensional distributions. Moreover, looking at the medians of the total variations, total degeneracy to delta-distributions occurs for more than half of the implicit distributions of $\mathcal{L}_\gamma^{\mathrm{h}}$ with $\gamma = 1.0$ (ELBO), $\mathcal{L}_\gamma^{\mathrm{bh}}$ with $\gamma = 0.1$ and $\mathcal{L}_\gamma^{\mathrm{bh}}$-alt with $\gamma = 0.9$. These agree with the plots in Fig. 6 and show that our approach cannot prevent degeneracies. Nonetheless, if we compare among the statistics for the $\mathcal{L}_\gamma^{\mathrm{h}}$s, we find that settings with $\gamma < 1$ give less degenerate distributions than ELBO ($\gamma = 1.0$).

The coefficient of variations are at least 1.9, indicating very different implicit distributions for the test samples. The skewness

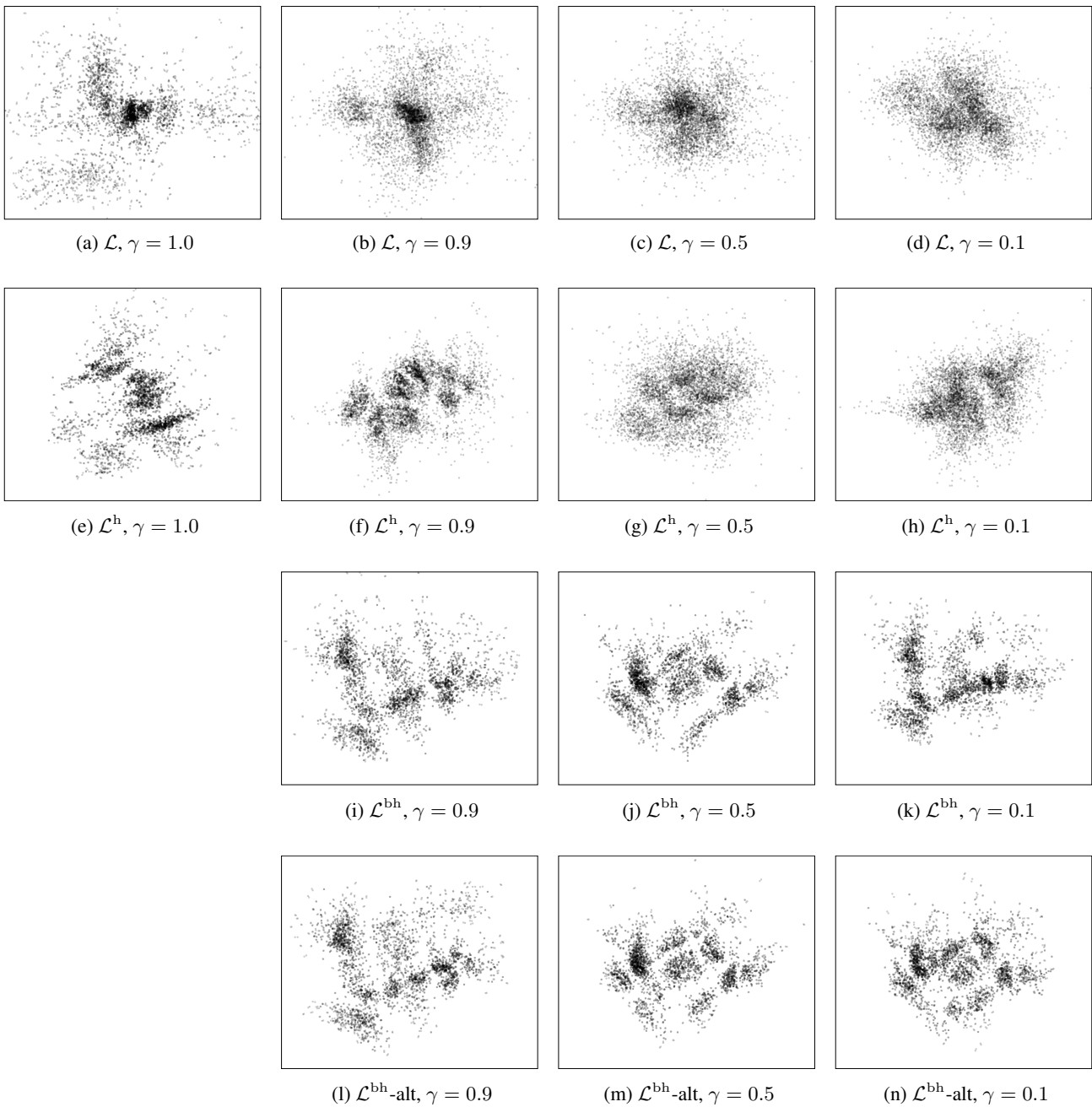

Figure 5: 5,000 samples from the posteriors of the MNIST test images. For $\mathcal{L}^{\text{bh}}$ and $\mathcal{L}^{\text{bh}}$-alt, the Bayes posteriors are used. All axes ranges from $-4$ to $4$.

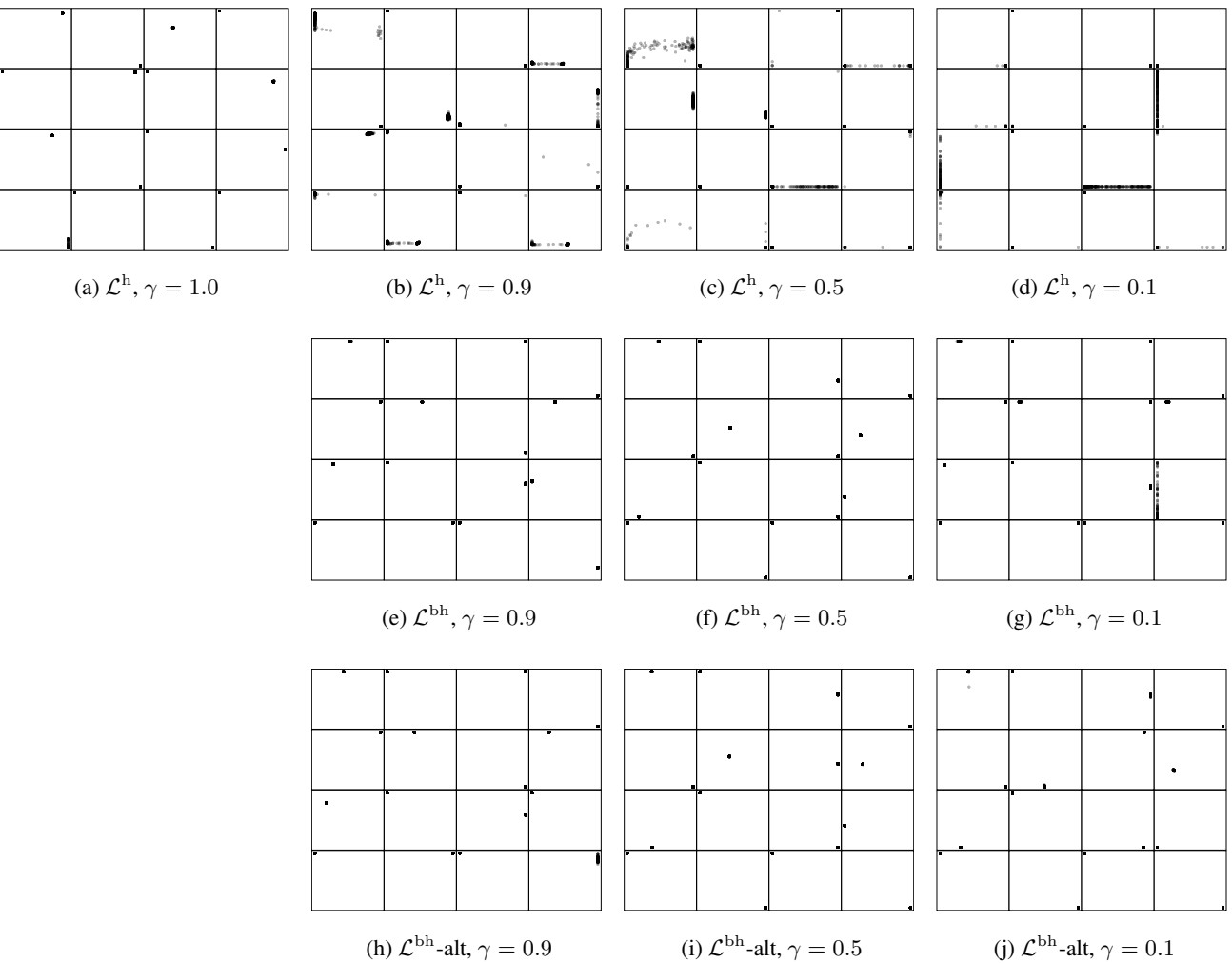

(a) $\mathcal{L}^{\text{h}}, \gamma = 1.0$     (b) $\mathcal{L}^{\text{h}}, \gamma = 0.9$     (c) $\mathcal{L}^{\text{h}}, \gamma = 0.5$     (d) $\mathcal{L}^{\text{h}}, \gamma = 0.1$

(e) $\mathcal{L}^{\text{bh}}, \gamma = 0.9$     (f) $\mathcal{L}^{\text{bh}}, \gamma = 0.5$     (g) $\mathcal{L}^{\text{bh}}, \gamma = 0.1$

(h) $\mathcal{L}^{\text{bh}}$-alt, $\gamma = 0.9$     (i) $\mathcal{L}^{\text{bh}}$-alt, $\gamma = 0.5$     (j) $\mathcal{L}^{\text{bh}}$-alt, $\gamma = 0.1$

Figure 6: 500 samples from the implicit posteriors for 16 MNIST test images: one small square is for one image. All axes ranges from 0 to 1, which is the range of the samples by design.

Table 8: Descriptive statistics, over the test images, of the generalised variance and total variation of the transformed implicit distributions. Figures less than $10^{-10}$ are treated as zero. For a number $a \times 10^{b}$, we express it as ${}_{a}b$, except that we use 0 when $a = 0$ and ${}_{a}$ when $b = 0$.

| Objective | $\gamma$ | Generalised Variance | | | | | Total Variation | | | | |
|---|---|---|---|---|---|---|---|---|---|---|---|
| | | Median | Max | Mean | CV | Skew | Median | Max | Mean | CV | Skew |
| $\mathcal{L}^{\mathrm{h}}$ | 1.0 | 0 | $_{5.4}{-3}$ | $_{3.2}{-6}$ | $_{3.0}1$ | $_{4.5}1$ | 0 | $_{8.2}{-1}$ | $_{2.0}{-3}$ | $_{1.1}1$ | $_{1.8}1$ |
| | 0.9 | $_{4.8}{-8}$ | $_{1.9}{-1}$ | $_{1.3}{-3}$ | 6.8 | $_{1.1}1$ | $_{3.0}{-3}$ | 1.1 | $_{5.5}{-2}$ | 2.3 | 3.6 |
| | 0.5 | 0 | $_{1.9}{-1}$ | $_{1.8}{-3}$ | 5.5 | 9.3 | $_{9.8}{-3}$ | 1.2 | $_{1.0}{-1}$ | 1.9 | 2.6 |
| | 0.1 | 0 | $_{1.5}{-1}$ | $_{1.3}{-3}$ | 6.4 | 9.7 | $_{3.8}{-5}$ | 1.0 | $_{5.9}{-2}$ | 2.3 | 2.9 |
| $\mathcal{L}^{\mathrm{bh}}$ | 0.9 | 0 | $_{1.1}{-2}$ | $_{1.5}{-6}$ | $_{7.4}1$ | $_{9.7}1$ | $_{4.5}{-9}$ | $_{4.5}{-1}$ | $_{1.3}{-3}$ | 9.9 | $_{1.6}1$ |
| | 0.5 | 0 | $_{1.5}{-2}$ | $_{2.5}{-6}$ | $_{6.3}1$ | $_{8.7}1$ | $_{3.9}{-8}$ | $_{4.8}{-1}$ | $_{1.7}{-3}$ | $_{1.0}1$ | $_{1.7}1$ |
| | 0.1 | 0 | $_{8.2}{-2}$ | $_{2.4}{-5}$ | $_{4.3}1$ | $_{6.6}1$ | 0 | $_{5.9}{-1}$ | $_{7.2}{-3}$ | 5.9 | 8.1 |
| $\mathcal{L}^{\mathrm{bh}}$-alt | 0.9 | 0 | $_{1.5}{-2}$ | $_{3.2}{-6}$ | $_{5.2}1$ | $_{7.9}1$ | 0 | $_{5.6}{-1}$ | $_{4.3}{-3}$ | 7.8 | $_{1.1}1$ |
| | 0.5 | 0 | $_{2.5}{-3}$ | $_{6.3}{-7}$ | $_{4.5}1$ | $_{7.4}1$ | $_{1.4}{-7}$ | $_{5.3}{-1}$ | $_{1.8}{-3}$ | $_{1.0}1$ | $_{1.7}1$ |
| | 0.1 | 0 | $_{7.4}{-3}$ | $_{4.1}{-6}$ | $_{3.2}1$ | $_{4.1}1$ | $_{3.0}{-6}$ | $_{5.6}{-1}$ | $_{4.2}{-3}$ | 7.4 | $_{1.1}1$ |

are positive, indicating high proportion of very low variance implicit distributions, and this is also shown by the medians being smaller than the means. In particular, the skewness for ELBO is about five times more than for $\mathcal{L}^{\mathrm{h}}_{\gamma}$ with $\gamma = 0.9$.

### C.5. Improving the VAE Decoder by Learning Fractional Posterior

Figures 7a to 7d provide sample images for the decoders trained with $\gamma$ taking values 1.0 (for ELBO), $10^{-1}$, $10^{-3}$ and $10^{-5}$. Visually, the best samples are provided by $\gamma = 10^{-5}$ (Fig. 7d). For decreasing $\gamma$, the FIDs are 83.5, 69.5, 67.8 and 68.8 (smaller is better). While the FIDs for the fractional posteriors are similar, they are all significantly better than the Bayes posterior's.

Since the $\beta$-VAE at its theoretical optimum also gives a power posterior, we also evaluate training with its objective, called $\mathcal{L}^{\beta}_{\beta}$. The fractional posterior for $\mathcal{L}^{\beta}_{\beta}$ corresponds directly to that for $\mathcal{L}_{\gamma}$ with $\beta = 1/\gamma$, so we use $10^{1}$, $10^{3}$ and $10^{5}$ for $\beta$. The FIDs in increasing $\beta$ are 77.3, 334.7 and 342.3. While the FIDs for $\mathcal{L}^{\beta}_{10}$ (corresponding to $\gamma = 10^{-1}$) improves over the 83.5 of ELBO's, it is significantly worse than the 69.5 of $\mathcal{L}_{10^{-1}}$. Moreover, increasing $\beta$ to $10^{3}$ and $10^{5}$ appears to cause degeneracy to a different and worse optima, in contrast to the stability afforded by $\mathcal{L}_{\gamma}$. Figures 7f to 7h provide the sample images. We further tried 5 and $10^{2}$ for $\beta$, giving FIDs 78.4 and 99.1.

Table 9 provides the statistics of the bounds to the data evidence for the trained VAEs. Similar to the results for MNIST (Table 2), $\mathcal{L}_{\gamma}$ with smaller $\gamma$ give tighter bounds and the learnt posteriors are closer to the prior. For $\mathcal{L}^{\beta}_{10}$, the bounds are looser than ELBO's, as expected. For $\mathcal{L}^{\beta}_{\beta}$ with $\beta \in \{10^{3}, 10^{5}\}$, the direct multiplication of the divergence term in $\mathcal{L}^{\beta}_{\beta}$ has cause instability such that the empirical divergence computed by sampling becomes negative — more samples than the 1,024 used here could resolve this issue for $\beta$-VAE.

All the preceding results are with two-dimensional latent space. We tested a case of four-dimensional latent space using our bound with $\gamma$ set to 1.0 (for ELBO), $10^{-3}$ and $10^{-5}$. With this more expressive model, the evidences are larger (last three row in Table 9), and the FIDs are better at 58.3, 56.8 and 55.8. Again, we see an advantage for $\gamma < 1.0$, though now the benefits are much less significant.

### C.6. Source Codes and Data Sets

Other than the standard Python and PyTorch (`https://pytorch.org/`), including Torchvision, packages, we take reference from and make use of the following source codes:

**VAE** `https://github.com/EmoryMLIP/DeepGenerativeModelingIntro`

**SIVI** `https://github.com/mingzhang-yin/SIVI`

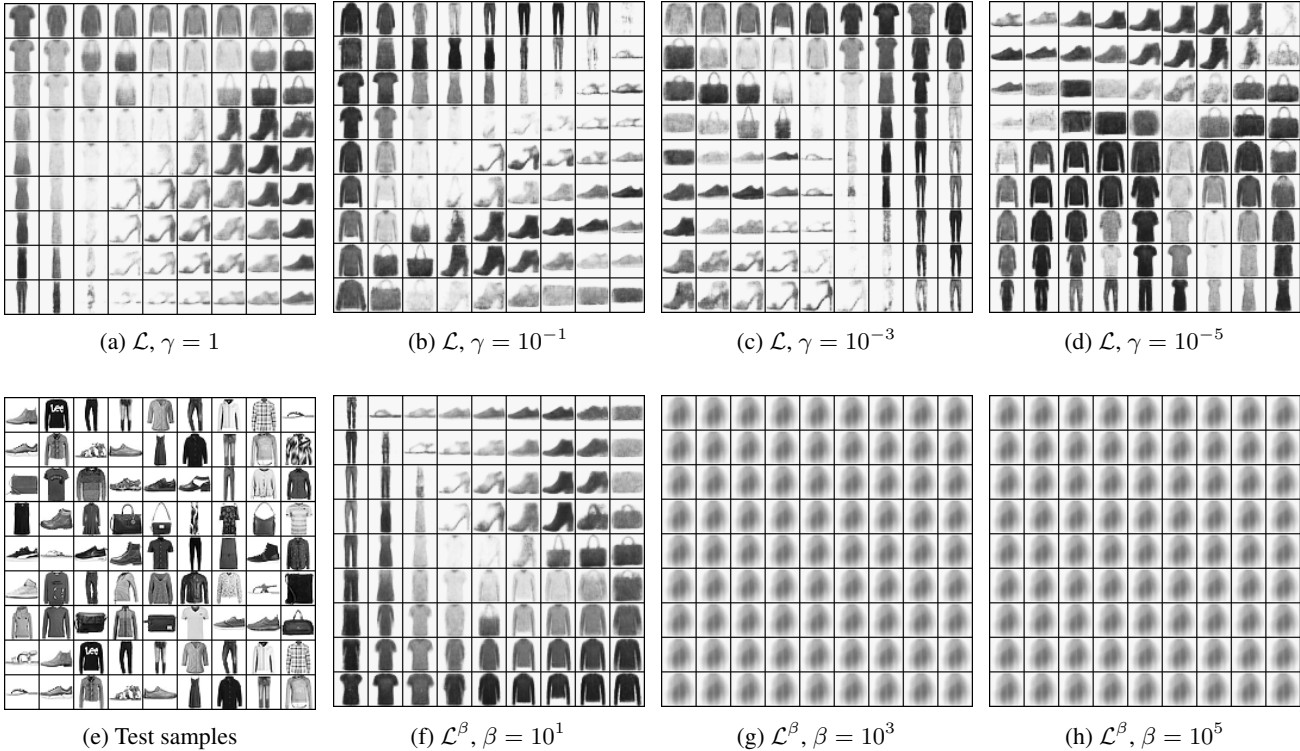

(a) $\mathcal{L}, \gamma = 1$     (b) $\mathcal{L}, \gamma = 10^{-1}$     (c) $\mathcal{L}, \gamma = 10^{-3}$     (d) $\mathcal{L}, \gamma = 10^{-5}$

(e) Test samples     (f) $\mathcal{L}^{\beta}, \beta = 10^{1}$     (g) $\mathcal{L}^{\beta}, \beta = 10^{3}$     (h) $\mathcal{L}^{\beta}, \beta = 10^{5}$

Figure 7: Mean images from decoded latent variables obtained by coordinate-wise inverse-CDF (standard normal) transform from a unit square. The VAE is trained on the Fashion-MNIST dataset. The top row uses the bound introduced in this paper; the bottom row (sans the first figure) uses the objective from $\beta$-VAE.

Table 9: Average log-evidences (higher better) over data samples, and its breakdown for VAE on Fashion-MNIST data sets. For Monte Carlo averages, 1,024 samples are used. For $\gamma = 1.0$, the figures are the same under *Test using Objective* and *Test using ELBO*. The columns under *Test using ELBO* are *solely* for diagnostics to understand the learnt posteriors using the same metrics: they are not performance measures. For ease of comparison, we also add the last column of FID scores of the 10,000 generated images.

| dim($z$) | Obj. | $\gamma$ or $\beta$ | Train (Total) | Test using Objective | | | Test using ELBO | | | FID |
|---|---|---|---|---|---|---|---|---|---|---|
| | | | | Total | data | div | Total | data | div | |
| 2 | $\mathcal{L}_{\gamma}$ | 1.0 | 1152.30 | 1118.15 | 1123.11 | 4.96 | 1118.15 | 1123.11 | 4.96 | 83.5 |
| | | $10^{-1}$ | 1180.08 | 1171.38 | 1173.97 | 2.59 | 902.79 | 904.56 | 1.77 | 69.5 |
| | | $10^{-3}$ | 1179.68 | 1171.05 | 1173.91 | 2.87 | 915.56 | 917.30 | 1.74 | 67.8 |
| | | $10^{-5}$ | 1183.97 | 1175.29 | 1178.04 | 2.75 | 853.45 | 855.16 | 1.71 | 68.8 |
| | $\mathcal{L}_{\beta}^{\beta}$ | 5.0 | 1135.83 | 1103.29 | 1121.04 | 17.76 | 1117.50 | 1121.05 | 3.55 | 78.4 |
| | | $10^{1}$ | 1120.62 | 1086.02 | 1117.13 | 31.11 | 1114.01 | 1117.12 | 3.11 | 77.3 |
| | | $10^{2}$ | 937.29 | 921.26 | 1067.22 | 145.95 | 1065.74 | 1067.20 | 1.46 | 99.1 |
| | | $10^{3}$ | 754.34 | 752.08 | 598.54 | $-153.54$ | 598.69 | 598.54 | $-0.15$ | 334.7 |
| | | $10^{5}$ | 15946.58 | 15952.71 | 598.53 | $-15354$ | 598.69 | 598.54 | $-0.15$ | 342.3 |
| 4 | $\mathcal{L}_{\gamma}$ | 1.0 | 1220.04 | 1200.40 | 1207.98 | 7.58 | 1200.40 | 1207.98 | 7.58 | 58.3 |
| | | $10^{-3}$ | 1231.16 | 1219.11 | 1225.14 | 6.04 | 1147.20 | 1151.29 | 4.09 | 56.8 |
| | | $10^{-5}$ | 1231.02 | 1219.24 | 1225.36 | 6.12 | 1147.87 | 1152.03 | 4.16 | 55.8 |

**FID** `https://github.com/mseitzer/pytorch-fid`

The above PyTorch code for FID is recommended by the original authors of FID at `https://github.com/bioinf-jku/TTUR`. Our source codes are available at `https://github.com/csiro-funml/Variational-learning-of-Fractional-Posteriors/`. They are executable on a single NVIDIA T4 GPU, which are available free (with limitations) on Google Collab, Kaggle and Amazon SageMaker Studio Lab at the point of writing. A single training run of the 500 epochs for the VAE experiments for $\mathcal{L}_\gamma$ is currently achievable within 12 hours on Kaggle.

The MNIST and Fashion-MNIST data sets are available via Torchvision.

