# OpenReview forum: "Variational Learning of Fractional Posteriors"
_ICML.cc/2025/Conference — ICML 2025 poster_

### Official Review · Reviewer_LKGu · 2025-03-06

**Overall Recommendation:** 3

**Summary:**

The authors introduce a novel variational inference method based on fractional posteriors, parameterized by a single scalar
γ∈(0,1). This approach generalizes Bayesian variational inference by tempering the likelihood term, leading to fractional posteriors that provide improved calibration and robustness. It extends to hierarchical models and shows applicability in both analytic and empirical scenarios, including Gaussian mixture models (GMMs) and variational autoencoders (VAEs). The main contributions include derivations of novel variational lower bounds based on Holder’s inequality, analytic gradient expressions, and empirical demonstrations showing improved calibration and generative modeling performance over conventional methods such as ELBO.

## update after rebuttal

I would like to thank the authors and the other reviewers for the discussion.
I keep my score, as explained in my comments.

**Claims And Evidence:**

Most claims are supported through theoretical derivations and by limited empirical results . The authors convincingly show the benefit of fractional posteriors for uncertainty calibration in GMMs and generative modeling with VAEs. However, some claims about robustness to model misspecification and improved alignment with priors are insufficiently supported by the empirical evaluations provided. Specifically, the paper claims robust performance improvement, but the supporting evidence is limited to simplified experiments that might not generalize to more complex scenarios.

**Essential References Not Discussed:**

Several relevant approaches are noticeably absent. Examples:

* Recent advances in "cold posteriors" (e.g., "How Good is the Bayes Posterior in Deep Neural Networks Really?" by Wenzel et al.) share conceptual similarities with fractional posteriors in controlling posterior calibration but are not discussed adequately.
* "A Simple Baseline for Bayesian Uncertainty in Deep Learning" by Maddox et al. which propose  Stochastic Weight Averaging (SWA) approach that also deal with tempered posteriors and could offer valuable comparisons.

Also many papers on mitigating posterior collapse are missing, such papers present an alternative to improving sample quality, especially when combined with learning an empirical prior post-training.

Examples:
* "Preventing Posterior Collapse with delta-VAEs" by Razavi et al.
* "beta-VAE: Learning Basic Visual Concepts with a Constrained Variational Framework" by Higgins et al.
* "Variational Lossy Autoencoder" by Chen et al.

Inclusion of such works would clarify the positioning of this paper's contributions relative to established posterior inference approaches and methods known for calibration.

**Experimental Designs Or Analyses:**

The empirical designs have notable issues. The calibration experiments with Gaussian mixtures provide insights into fractional posteriors' theoretical properties; however, only two-component mixtures are explored, which significantly limits the generalizability of results to more complex, realistic scenarios. The linear regressions used to select optimal γ values (Rℓ and Rκ) lack sufficient theoretical justification—why linear regression should capture the relationship between coverage intervals and fractional parameters?

In VAE experiments, it is unclear whether the observed improvements in generative quality generalize beyond the simple MNIST and Fashion-MNIST datasets, or the limited dimensionality (two-dimensional latent space). Experiments involving higher-dimensional latent spaces or more complex datasets are needed to substantiate the broader claims about generative performance improvements.

**Methods And Evaluation Criteria:**

The methods and evaluation criteria are mostly appropriate. The use of simple benchmarks like MNIST and Gaussian mixtures is effective for demonstrating conceptual advantages, though it somewhat limits the impact. For the calibration study on the GMM, the chosen metric (credible intervals and coverage probabilities) is appropriate. However, broader applicability could have been shown with additional, more complex datasets or larger-scale problems to demonstrate practical relevance clearly.

**Other Comments Or Suggestions:**

* Figure 1 - clarity could be improved by adding explanation to each of the labels (a-d) explicitly. (b,c) looks a lot like latent interpolation, is that the case?
* Table 2 - captions and discussions should explicitly clarify the implications of differences between empirical evidence bounds and variational approximations. If "Test using ELBO are solely for diagnostics to understand the learnt posteriors using the same metrics" then can you please explain what do we learn from it or alternatively remove it?
* Line 069 - should be "(KL, α → 1)" and not "(KL, α = 1)"
* Line 252 - what is the error for the MC estimates of Zc and Zd? Can you provide any bound? How will large error affect the learned parameters of a VAE model?

**Other Strengths And Weaknesses:**

Strengths:
* Clearly derived and theoretically justified new variational lower bounds.
* Novelty in combining fractional posterior methods with variational inference.
* Empirical demonstration that fractional posteriors improve generative tasks in VAEs.

Weaknesses:
* Experiments limited in complexity (simple datasets, low-dimensional settings).
* Lack of comprehensive evaluation on scalability or applicability to large-scale practical tasks.
* Limited sensitivity analysis regarding the γ hyperparameter.

**Questions For Authors:**

* How sensitive are results to different choices of γ? Would a systematic exploration (beyond the limited set used) reveal significantly different findings regarding calibration or performance?

* Why did you not test your approach on higher-dimensional latent spaces or more complex data? Would you expect similar benefits in such scenarios, and if not, why?

* Can you clarify the conditions under which the introduced hierarchical approach (section 2.3) avoids the degeneracy problem of ELBO? An explicit analysis would substantially strengthen the theoretical contribution.

* The various approximations might limit the practicality of the proposed method to simpler dataset and lower dimensions. Have you tested the MC approximation error for latent space with more dimensions and larger and more complex images?

* What is the advantage of the proposed method compared to many other methods that address posterior collapse in VAE?

* Learning an empirical prior after training can also provide high quality samples (assuming posterior collapse was mitigated) and might be a simpler and a more practical solution. Can you explain why would the proposed method be beneficial here?

* In  section C.4 clearly demonstrates areas of the prior (explored via interpolation) where the samples are noisy (Figures 3,6).
Given the very simple datasets (MNIST, fashion-MNIST), it is not clear how well the advantages of the proposed method will scale to more complex and higher dimensional data. Have you tried the proposed method with such datasets (e.g., ImageNet or even CIFAR10)?

**Relation To Broader Scientific Literature:**

The paper situates its contributions clearly within the context of existing literature on fractional posteriors, variational inference (VI), and related methods (e.g., β-VAE, importance-weighted ELBO, and general fractional posterior methods). It explicitly connects its novel bounds with existing concepts like PAC-Bayes and generalized variational inference (Knoblauch et al., 2022). However, the paper downplays related work that explores tempering or regularization in posterior inference (e.g., mitigating posterior collapse), which could contextualize the novelty more comprehensively.

**Theoretical Claims:**

The main theoretical derivations involving the lower bounds via Hölder’s inequality (Equations (1) and (2)) appear mathematically sound. However, the proof that the fractional posterior exactly corresponds to an optimal lower bound (Section 2.1) should be elaborated further to highlight clearly any assumptions required. Additionally, the limits as γ approaches 1 (recovering ELBO) are adequately proven, though readers might benefit from more intuitive explanations (which are indeed present in the supplementary materials).

---

> ### Author Rebuttal · Authors · 2025-03-30
>
> > some claims … insufficiently supported by the empirical
>
> Lines 12 (right) and 87 (right) are general remarks on robustness of fractional posteriors, citing others. Statement on line 44 (right) follows these, but may be misunderstood. We will change to “an alternative to”.
>
> Alignment with prior is substantiated by Table 2 (last col), Fig 4&5, and  Fig 7 & Table 5.
>
> ### Experimental
> > only two-component mixtures
>
> Beyond 2 components, the marginal likelihood landscape is complex (line 328, left, citation therein). The main text uses 2 components for clarity and focus. Beyond this, we need more exposition to separate the confounding effects due to the complex landscape. Nonetheless, C.1.2 gives results for 4 components.
>
> > linear regressions … lack sufficient theoretical justification
>
> Indeed, it’s more justified to regress against $\ell^{-2}$ (C.1.1). This gives better interval lengths, but worse coverages. Other strategies are possible (last sent. in sec. 6).
>
> ### Comments (numbered)
> 1. Yes. See caption (also Fig 4 of Kingma&Welling, 2022]. marginal-CDF$\in [0.1, 0.9]$
> 2. All results are based on the variational approximation optimised on train set. The empirical error-bars are for 10 runs, a different seed each to initialise the NN.
>
> With “ELBO/div”, we know that the fractional posteriors are closer to the prior for smaller $\gamma$ in the same sense of KL. This can't be seen from “Objective/div” because Renyi-divergence differs with $\gamma$. See also line 358 (left)
>
> 3. Ok
> 4. We haven't systematically investigated. Table 2 hints at stability from the relatively small error bars (3 x stddev) by using 100 samples per datum and 10 runs (details in C.3).
>
> ### Questions (numbered)
> 1. Sensitivity analysis of $\gamma$ must be done within posterior families (sec 2.2). Current results show the quality of the bound and the closeness of the approx. fractional posterior to the prior.
>
> 2. We will acknowledge this experimental limitation in an additional section.
>
> Current 2d latent space allows posterior illustrations (Fig 4 to 6). We use NNs from [Ruthotto & Haber, 2021], with only 88,837 parameters for 2d latent space, so results can be obtained on free platforms (sec C.5). We need richer NNs for more complex data.
>
> We are also careful and use the correct likelihood: continuous Bernoulli. Otherwise, bounds in Table 2 are not convincing.
>
> We expect similar benefits compared with existing objectives such as ELBO or $\beta$-VAE’s, when the goal is either better evidence bounds, or more stable computation of fractional posteriors, or both.
>
> 3. Degeneracy isn't necessary for optimality; but it isn't avoided and is also a solution, as stated in A.4. There, Eqs 3&4 say if we explicitly enforce $q(u)\neq0$ at multiple locations, optimality is still possible. So, we can design $q(u)$ to be non-degenerate if we wish. Further work can be on crafting training dynamics towards non-degeneracy.
>
> 4. For FashionMNIST, using 4d latent space and the following $\gamma$s for our primary bound:
> * 1 (ELBO): train/test bound=1220/1200; FID=58.3
> * $10^{-3}$: train/test bound=1231/1219; FID=56.8
> * $10^{-5}$: train/test bound=1231/1219; FID=55.8
>
> So for 4d, bound for smaller $\gamma$ gives better results. Overall 4d results better than 2d (see C.4 for existing values), as expected for a richer model with more parameters.
>
> We haven't analysed the MC approx error.
>
> 5. We don't directly address posterior collapse. If we must, we might seem to be encouraging collapse, but it's more complicated and demands more discussion. Figure 4(d) gives a prelude, where the fractional posteriors as a whole aggregate towards the prior, but posterior for every data point is different.
>
> 6. With respect to image generation from prior (sec 5.3), we learn a decoder that operates well with the prior, which can be simple. At the same time and within the same objective, the NN are optimised with respect to the fractional posteriors that are close to the prior.
>
> Post learning the empirical prior requires its model to be sufficiently flexible to approx the posteriors, and a separate procedure to learn its parameters.
>
> So we believe our proposal is simpler.
>
> Which is more practical depends on context within a wider ML/AI sys — e.g., if posteriors exist from previous intensive training/tuning, the empirical prior approach might be preferred.
>
> 7. The samples are noisy also because NN architectures for the encoder & decoder are simple with 89K parameters for 2d latent space. For CIFAR10, we add one CNN layer (total 3) each for encoder & decoder, total 530K parameters for 32d latent space. Number of train/eval samples reduced to16/128; epochs reduced to 300 (C.3 gives current settings). Results using SageMaker StudioLab:
> * $\gamma=1$ (ELBO): train/test bound=1229/1172; FID=141
> * $\gamma=10^{-5}$: train/test bound=1238/1184; FID=135
>
> Our bound for $10^{-5}$-posterior gives better results. Better results need more complex architectures, e.g., ResNet-1001 with 10.2M parameters.

---

> > ### Comment · Reviewer_LKGu · 2025-04-01
> >
> > Thanks for the detailed rebuttal. I will keep the score.

---

### Official Review · Reviewer_SAc3 · 2025-03-07

**Overall Recommendation:** 4

**Summary:**

The paper introduces a novel one-parameter variational objective that generalizes the standard evidence lower bound (ELBO) by enabling the estimation of fractional posteriors. By leveraging Hölder’s inequality, the authors derive a new bound L₍γ₎ which recovers the conventional ELBO in the limit as γ → 1. The framework is further extended to hierarchical and Bayes posteriors, and the paper provides both analytical gradient derivations for cases such as exponential family models and mixture models, as well as empirical studies. Experiments on Gaussian mixture models and variational autoencoders (VAEs) demonstrate that fractional posteriors yield better-calibrated uncertainties and improve generative performance, particularly in aligning the VAE decoder with the prior distribution.

**Updates after Rebuttal**

Thanks for the rebuttal. I think most of my concerns have been addressed. I will keep my scores.

**Claims And Evidence:**

Yes all claims are supported by clear theoretical and empirical evidence

**Essential References Not Discussed:**

The paper covered most of the essential works in this field.

**Experimental Designs Or Analyses:**

Yes I have checked the soundness and validity of experimental designs, and they are sound.

**Methods And Evaluation Criteria:**

Yes

**Other Comments Or Suggestions:**

N.A.

**Other Strengths And Weaknesses:**

Strengths:
* The experiments on VAEs, including improvements in evidence bounds and better alignment of decoder distributions for generative tasks, indicate potential practical benefits in generative modeling and beyond.

Weaknesses
* The method involves nested integrations and Monte Carlo estimates (especially in the hierarchical and semi-implicit formulations), which may lead to high computational overhead. This can become particularly problematic in high-dimensional latent spaces or when dealing with complex models.

**Questions For Authors:**

How would the proposed fractional posterior works when a generation framework other than VAE is applied? Say a flow-based framework.

**Relation To Broader Scientific Literature:**

The proposed approach is closely linked to the density estimation literature by tempering the likelihood to improve robustness and convergence properties. For instance, Friel and Pettitt (2008)  introduced power posteriors to facilitate marginal likelihood estimation in mixture models—a concept that underpins the use of fractional likelihoods in density estimation. Similarly, O’Hagan (1995)
ARXIV.ORG
 showed that raising the likelihood to a fractional power can yield more objective Bayes factors, particularly valuable in nonparametric settings where standard models are prone to overfitting.

In the context of variational autoencoders (Kingma & Welling, 2014), the paper’s findings relate to efforts such as β-VAE (Higgins et al., 2017) and IWAE (Domke & Sheldon, 2018), where modifying the ELBO has been proposed to improve disentanglement or tighten the variational bound.

**Theoretical Claims:**

I have checked all theoretical proofs, and I found no overt mistakes in the proofs.

---

> ### Author Rebuttal · Authors · 2025-03-28
>
> ### Relation To Broader Scientific Literature
>
> Thank you. We will try our best to incorporate these into the paper.
>
> ### Other Strengths And Weaknesses
> > The method involves nested integrations and Monte Carlo estimates (especially in the hierarchical and semi-implicit formulations), which may lead to high computational overhead. This can become particularly problematic in high-dimensional latent spaces or when dealing with complex models.
>
> As in our reply to Reviewer H3En, the triple integral in section 2.3.2 can be reduced to a double integral (see section A.3 and last row of Table 3). The remaining double integrals are necessary since we have a hierarchical construction, involving $\boldsymbol{u}$ and then $\boldsymbol{z}|\boldsymbol{u}$. Fortunately, depending on the application, one may not need such a construction if we set $\gamma$ to be small so that the approximating family can be simple, as discussed in section 2.2.
>
> ### Questions For Authors
> We have done preliminary work on such extensions, but it is not trivial. We leave such matters for future work to keep the current paper focused. Subject to space constraints, we may suggest approaches in relation to other frameworks in an additional future work or discussion section, especially on pitfalls to avoid based on our experience.

---

> > ### Comment · Reviewer_SAc3 · 2025-04-02
> >
> > Thanks for the rebuttal. I think most of my concerns have been addressed. I will keep my scores.

---

### Official Review · Reviewer_F6u2 · 2025-03-09

**Overall Recommendation:** 3

**Summary:**

The paper proposes a variational objective targetting fractional posteriors based on the Holder inequality instead of the more typical Jensen's inequality approach. Furthermore, for hierarchical models, minor variations of the variational objective are considered. The utility of the approach is demonstrated through coordinate-ascent variational inference of mixture models and gradient-based variational expectation maximization of deep latent variable models.

## update after rebuttal
Through intense discussion with the authors, the points of disagreement have been sufficiently identified, and the authors have promised to address them. As such, I am now in favor of the paper to be accepted. However, the rather large number of changes promised by the authors makes me think that an additional round of review would be beneficial. As such, I will lean toward borderline.

**Claims And Evidence:**

The contributions of the paper are a bit confounded, and this is my main concern. The main claim appears to be that the paper proposes a variational objective that can handle fractional posteriors. What I don't get is that one can use plain-old evidence lower bound maximization for fractional posteriors too: what is wrong with using the ELBO
$$
\mathcal{L}_{\mathrm{ELBO}}^{\gamma}(q) = \int \big\\{ \gamma \log p(\mathcal{D} \mid z) + \log p(z) \big\\} q(\mathrm{d}z) + \mathbb{H}(q)
$$
for approximating a $\gamma$-fractional posterior?
Therefore, I do not find that being able to handle fractional posteriors can be claimed as a technical contribution.

On the other hand, it does appear that the paper is proposing a novel variational objective based on Holder's inequality. But it is unclear what the claimed benefits of this new objective are. Is this statistically better in any sense than naively using the ELBO as above? Is there a computational benefit? Works that propose new divergences need to articulate what is uniquely new or desirable about the proposed divergence. For instance, the initial claim about $\alpha$- and $\chi^2$-divergences [1,2] was that they reduce the mode-seeking behavior of the exclusive KL divergence. The long-standing argument for the exclusive KL divergence is that it is computationally convenient to optimize.

Unless there is some serious misunderstanding on my end, I think the paper should reconsider its positioning and refine its technical claims.

1. Li, Yingzhen, and Richard E. Turner. "Rényi divergence variational inference." Advances in neural information processing systems 29 (2016).
2. Dieng, Adji Bousso, et al. "Variational Inference via $\chi $ Upper Bound Minimization." Advances in Neural Information Processing Systems 30 (2017).

**Essential References Not Discussed:**

* Section 2.3: The objectives specialized to hierarchical models in Section 2.3 are reminiscent of the "locally-enhanced bounds" proposed in [1]. I recommend taking a look for connections.
* Line 228: The reparameterization gradient for variational inference was independently proposed by [2,3] as well.

1. Geffner, Tomas, and Justin Domke. "Variational inference with locally enhanced bounds for hierarchical models." arXiv preprint arXiv:2203.04432 (2022).
2. Titsias, Michalis, and Miguel Lázaro-Gredilla. "Doubly stochastic variational Bayes for non-conjugate inference." International conference on machine learning. PMLR, 2014.
3. Rezende, Danilo Jimenez, Shakir Mohamed, and Daan Wierstra. "Stochastic backpropagation and approximate inference in deep generative models." International conference on machine learning. PMLR, 2014.

**Experimental Designs Or Analyses:**

For reasons mentioned above, the evaluation is not entirely sound. The paper is proposing a variational objective and corresponding inference algorithms. Thus, the evaluation should focus on evaluating the following: "Given a fixed value of $\gamma$, how accurate is the obtained variational approximation?" The experiments are not appropriate for doing this.

The experiments in Table 1, for example, are not doing this: coverage confounds the properties of the model and the algorithm, so they are not informative about the performance of the algorithm/variational objective. Furthermore, since the model changes depending on $\gamma$, the variational bound values in between different values of $\gamma$ cannot be compared, and individual bound values are not interpretable. Similar arguments apply to Table 2.

**Methods And Evaluation Criteria:**

The same comments apply here. On a technical level, however, I would also like to point out that it is unclear what the objective $\mathcal{L}_{\gamma}$ is doing. It is fair to assume that it is a surrogate for some divergence, but which one? Is the proposed objective equivalent to this divergence up to a constant (The ELBO is exactly the same as the KL divergence up to a constant), or is it a strict surrogate?

Furthermore, the discussion in Section 2.2 needs to be more subtle. Selecting $\gamma$ is a specification of the *model*, not the *inference algorithm*. That is, the submission claims that if a variational approximation can "approximate only certain fractional posteriors well, then the corresponding $\gamma$s would be optimal." Optimal in what sense? This is essentially modifying the *model* so that some variational approximation better approximates it, but there is no reason to believe that this will be statistically sound. The model that is best approximated could be terrible. Instead, previous works took different approaches, from maximizing the predictive density [1] or taking the ABC perspective [2], which come with accompanying theoretical analyses.

1. Grünwald, Peter, and Thijs Van Ommen. "Inconsistency of Bayesian inference for misspecified linear models, and a proposal for repairing it." Bayesian Analysis (2017): 1069-1103.
2. Miller, Jeffrey W., and David B. Dunson. "Robust Bayesian inference via coarsening." Journal of the American Statistical Association (2019).

**Other Comments Or Suggestions:**

* The term "hierarchical posterior" has been used throughout. This is a bit unusual as a posterior may not have any notion of hierarchy, but a *model* can be hierarchical.
* Line 30 "high-variance estimators": Why is Roeder 2017 cited here? Roeder proposes a lower variance estimator, so this doesn't appear to be the right citation here. Furthermore, the fact that coordinate descent can be used does not necessarily fix everything since it can only be used with mean-field conjugate families.
* Line 69-72 "The Kullback-Leibler divergence ... is the only case where the chain rule of conditional probability": I am not sure why this is relevant here.
* Line 94 "it is achieved without relying on PAC-Bayes or modifying the likelihood": PAC-Bayes bounds are generalization bounds and not about variational inference algorithms. (Alquier *et al.* 2016 establish generalization guarantees for variational posteriors.) What did the authors intend by this sentence?

**Other Strengths And Weaknesses:**

n/a

**Questions For Authors:**

n/a

**Relation To Broader Scientific Literature:**

The paper proposes a new variational objective, which extends the existing literature on developing alternatives to the ELBOs. Given the concerns above, it is unclear how the work is positioning itself within this context.

**Theoretical Claims:**

n/a

---

> ### Author Rebuttal · Authors · 2025-03-28
>
> We suspect misunderstandings. We ask for further clarifications below.
>
> ### Claims & Evidence
> > contributions ... confounded
>
> Our key contribution is a lower bound that _also_ approximates fractional posteriors. Having both at once is new in ML and statistics. Moreover, ELBO is a special case. So, we fill a gap between standard VI (ELBO) and fractional Bayesian inference.
>
> Fractional posteriors have been shown to be more robust (sec 1 2nd para), and have applications in calibration (sec 5.1).  A lower bound allows _principled_ maximisation of the hyperparameters (ML-II), common in Bayesian ML (sec 5.2). Having both at once benefits generation from VAE decoder (sec 5.3).
>
> We bring the bound in sec 2 to expts in sec 5 by developing complex posterior constructions (sec 2.3), parameter updates (sec 3) and MC estimates (sec 4).
>
> > $L_{ELBO}^{\gamma}$
>
> $L_{ELBO}^{\gamma}$ essentially changes the likelihood of the model, so it is an ELBO to that changed model, and may not lower bound the original model (depends whether $p(D|z)>1$). In contrast, ours is a theoretical lower bound to the original model. It holds for all $\gamma\in(0,1)$ and all $q$.
>
> Nonetheless, the optimal solution of $L_{ELBO}^{\gamma}$ indeed approximates the fractional posterior of the original model. It's the same as $\beta$-VAE’s objective, by dividing by $\gamma$ (Table 3). If we only want a fractional posterior, then this suffices. If, in addition, we wish to estimate the hyperparameters of the model, as is common in ML, then our lower bound is effective (Table 2).
>
> > statistically better in any sense than naively using the ELBO ...?
>
> We show our bound to be computationally more stable and to give better results than $\beta$-VAE for the FashionMNIST in C.4 (referenced in sec 6). We have not analysed the properties such as convergence rates. Please clarify if we misunderstood the question.
>
> > computational benefit?
>
> No. Indeed, KL is mathematically more convenient. Computationally, if implemented using sampling (e.g., for VAE), we swap the log and sum operations, include multiplicative and additive constants, and use more than one sample (line 807).
>
> > Works that propose new divergences …what is uniquely new or desirable about the proposed divergence
>
> We provide a variational objective that lower bounds evidence _and_ enables estimation of approx. fractional posteriors. This is _not_ fulfilled by $L_{ELBO}^{\gamma}$ (nor $\beta$-VAE).
>
> ### Methods & Evaluation
> > ... a surrogate for some divergence, but which one? ... equivalent to this divergence up to a constant ... a strict surrogate?
>
> Optimality is at the fractional posterior so it's not a divergence to/from the Bayes posterior. We do not think it’s equivalent to any known divergence up to a constant (the evidence). We can define a divergence (from the fractional posterior) based on our bound: $D(q||(p(z),p(D|z),\gamma)) = L_{evd} - L_\gamma$, where the triple defines the target posterior.
>
> We can better answer if “strict surrogate” is defined.
>
> > $\gamma$ is a specification of the model, not the inference
>
> $\gamma$ is not a specification of the model. From a Bayesian viewpoint, the model is fully specified by likelihood $p(D|z)$ and prior $p(z)$. Given a data set, this fixes the exact marginal likelihood or data-evidence $p(D)$ (MacKay, 2003).
>
> One may compute $p(D)$ and $p(z|D)$ using sampling, such as MCMC. For complex models in ML, variational methods are developed (see 1st para of paper). We propose a new variational method that leads to approximate fractional posteriors. Our method has a parameter $\gamma$ for the inference, but it does not change the model.
>
> We admit there are schools other than Bayesian. In particular, in the regularisation community, the end objective is treated as the model, like the reviewer alludes to. To put our paper in the correct setting, it begins with “Exact Bayesian inference is ...”.
>
> > Optimal in what sense?
>
> In the sense of giving a tighter bound to the evidence (1st sent in para). We will reiterate at the end of para.
>
> > modifying the model … statistically sound.
>
> We emphasise that the model is fixed (given hyperparameters) and evidence is fixed (given data) in the Bayesian view. Our lower bound to the evidence is mathematically sound. It changes neither $p(D|z)$ nor $p(z)$; max wrt $q$ also changes neither. But max wrt $p(D|z)$ or $p(z)$ changes the model.
>
> ### Experiments
> Our evaluation is sound: **In sec 5.1, the model is fixed** and the bounds are comparable for relative tightness to the same exact evidence. **In sec 5.2, the model changes** because we optimise the decoder, changing $p(D|z)$ (right para starting line 318). Here, the bounds are comparable for the quality of optimised models within the same family (i.e., same NN architecture) on the same data.
>
> We should have used the word “tighter” more carefully. We will update the abstract; 4th para in intro; last para in sec 2.2; intro to sec 5. 4th and 5th para in sec 5.2; and conclusion.

---

> > ### Comment · Reviewer_F6u2 · 2025-04-01
> >
> > Thank you for the detailed response. I think I have a clearer understanding of the disagreements here.
> >
> > > $\gamma$ is not a specification of the model.
> >
> > Okay let use more clear terms; it is a specification of the *posterior*. Changing $\gamma$ changes the posterior. Now, if you are being orthodox Bayesian, there is no reason to change the posterior. The whole point of using fractional posteriors, however, is that you suspect the model is misspecified and therefore use modify the posterior.
> >
> > > $L_{ELBO}^{\gamma}$ essentially changes the likelihood of the model, so it is an ELBO to that changed model, and may not lower bound the original model (depends whether). In contrast, ours is a theoretical lower bound to the original model. It holds for all $\gamma \in (0, 1)$ and all $q$.
> >
> > I think the authors should clarify whether they are thinking in terms of performing empirical Bayes/marginal likelihood maximization (optimizing parameters of the posterior) or variational inference (inferring $q$). In the perspective of VI, it doesn't matter whether there exists an upper bound or not, what matters is which divergence is being minimized with respect to what target. Here, the target is the fractional posterior, which changes with $\gamma$.
> >
> > In terms of marginal likelihood maximization, it is unusual that variational inference is performed against the fractional posterior, but the marginal likelihood is defined with the original posterior. If one believes in their model, it makes sense to do everything with the untempered posterior. If you're being post-Bayesian and don't believe your posterior, then everything should be done using the fractional posterior, and this could be done by maximizing the $L^{\gamma}_{\mathrm{ELBO}}$. What is the justification for this mix and match here?
> >
> > > We provide a variational objective that lower bounds evidence and enables estimation of approx. fractional posteriors. This is not fulfilled by
> >
> > By "confounding," I meant that those two things need to be evaluated separately.
> >
> > > We can define a divergence (from the fractional posterior) based on our bound
> >
> > Yes please define this divergence so that it is clear what the variational inference part is actually doing.
> >
> > > In the sense of giving a tighter bound to the evidence (1st sent in para). We will reiterate at the end of para.
> >
> > In terms of marginal likelihood maximization, I can buy this. However, it must also be clarified that the variational approximation $q$ is now targetting a fractional posterior, where its $\gamma$ was not selected according to any notion of statistical optimality of the posterior but just to make the variational bound tighter.
> >
> > Here is a summary of the points:
> > * If marginal likelihood maximization is the only ultimate goal, then I agree that the technique in the paper could yield a tighter bound. I agree that this is meaningful.
> > * The technique proposed in this paper has little to do with the goal of post-Bayesian works in fractional posteriors since $\gamma$ is selected purely to make the variational bound tighter. Therefore, there is no reason that the usual goals of post-Bayesianism, like robustness against misspecification or calibration, will be fulfilled. This should be clarified in the paper and the evaluation will also have to reflect this.
> >
> > Do the authors concur with these points?

---

> > > ### Author Response · Authors · 2025-04-04
> > >
> > > > orthodox Bayesian … no reason to change the posterior. The whole point of using fractional posteriors … model is misspecified
> > >
> > > Agree. Also, the approx Bayes posterior by ELBO is known to under-estimate the uncertainty, so here we actually want an approx fractional posterior (sec 1 para 2). Sec 5.1 is on this.
> > > > clarify whether … empirical Bayes/marginal likelihood maximization (optimizing parameters of the posterior) or variational inference (inferring $q$)
> > >
> > > We assume the 1st parenthesis should be about “the model”.
> > >
> > > Until and including sec 5.1, the paper is on inferring $q$ to max the lower bound to the evidence. In sec 3, $\theta$ is the parameters of $\tilde{q}$; and sec 4 follows from this.
> > >
> > > Sec 5.2 & 5.3 optimise the hyperparameters in $p(D|z)$ of the model, as we feel is expected in VAE expts. Sec 5.2 para 5 makes this clear. We will add more signposting, and also use the word “tighter” more carefully.
> > > > doesn't matter ... an upper bound or not, what matters is which divergence is being minimized with respect to what target. Here, the target is the fractional posterior, which changes with $\gamma$
> > >
> > > Taking reference from [Knoblauch et al., 2022] cited in our paper, we feel the reviewer has taken the _VI as constrained optimization_ view (sec 2.3.3 therein), while we take the _VI as log evidence bound_ view (sec 2.3.1 therein). We acknowledge both. See citations in sec 6 para 1 and in sec 1 para 1.
> > > > marginal likelihood is defined with the original posterior
> > >
> > > Marginal likelihood is wrt the prior.
> > > > post-Bayesian and don't believe your posterior, then everything … done using the fractional posterior, … done by maximizing the $L_{ELBO}^\gamma$
> > >
> > > Yes, $L_{ELBO}^\gamma$ is a straightforward manner to achieve this. It obtains the same result as our bound if the approximating family includes the exact fractional posterior. If the approximating family does not include the exact posterior, then our bound also provides a meaningful quantification between different families and the optimal results therein. In the case of ELBO, access to such quantification has advanced ML, e.g., [Geffner and Domke, ICML 2022] cited by the reviewer.
> > > > justification for this mix and match
> > > * In sec 3.1.2, we can analyse the fractional posterior which approaches the Bayes posterior as the dataset grows (so $\gamma\rightarrow1$).
> > > * In sec 3.1.3, we use approximate fractional posterior for the component means and approximate Bayes posterior for the cluster assignment. Admittedly, we can also do this with some version of $L_{ELBO}^\gamma$, but in general the optimal posteriors will be different.
> > > * While the focus in sec 5.1 is on calibration, the bounds could also be used for model comparison against a different prior for the component means — this we have not done to maintain the focus of the section.
> > > * Sec 5.3 gives the case for learning VAE decoder so that we can generate via the prior.
> > >
> > > Our work may inspire more applications (see reply to Reviewer SAc3).
> > > > those two things … evaluated separately.
> > >
> > > Sec 5.1 gives both tightness of the bounds and the quality of the fractional posteriors. While the results are from the same experiment, we _assess the fractional posterior and the bounds separately_. We haven’t drawn conclusions from one to the other — e.g., we haven’t said that because it’s a fractional posterior, it’s a better bound. _Sec 2.2 forbids this explicitly_. We will restate this in sec 5.1. In fact Table 4 shows that smaller $\gamma$ can give worse bounds. We will highlight this in a para on bounds in sec 5.1 that was removed for space. We can also include the actual evidences for the simulated data for comparison.
> > >
> > > The same can be said for sec 5.2, though now the bounds are for the quality of the optimised models. We will reiterate that the two aspects are evaluated separately.
> > > > define this divergence …
> > >
> > > Will put this between sec 2.1 and 2.2.
> > > > $q$ is now targeting a fractional posterior … not selected according to any notion of statistical optimality of the posterior but just to make the variational bound tighter.
> > >
> > > This will be made explicit in sec 5.2.
> > >
> > > We add that sec 5.3 is a case for $\gamma$ to be small, though not for statistical optimality.
> > > > Here is a summary of the points:
> > > * Yes
> > > * Post-Bayesian has many aspects; we only do fractional posteriors. We cite existing work on robustness of fractional posteriors for context — _our work is not about proving robustness_ (see reply to Reviewer LKGu). We will make this clearer.
> > >
> > >     Sec 5.1 is on misspecification caused by variational inference (sec 6 last para; also Knoblauch et al., 2022 sec 2.4 pt ii). Some may disagree that this is misspecification — we can add this qualification. Here, $\gamma$ is chosen for calibration, not bounds.
> > >
> > >     An additional limitations sec can say
> > >     1. We haven't evaluated where either the likelihood or the prior is misspecified.
> > >     2. We don't directly address the goals of post-Bayesianism. We rely on the works of others on such matters.

---

### Official Review · Reviewer_H3En · 2025-03-09

**Overall Recommendation:** 3

**Summary:**

The classical variational inference (VI) often underestimates the uncertainty, motivating the research of generalized Bayesian inference. Nevertheless, the theoretical connections between generalized Bayesian inference bounds and the marginal likelihoods are only established approximately/asymptotically, hindering careful use in practice. This work shows that a family of generalized Bayesian inference bounds are lower bounds to the log marginals with an application of the Hölder’s inequality. The bound is tight when the approximate posterior is a fractional posterior, which interpolates between the true posterior and the prior. There are two main applications of the bound. The calibration of the approximate posterior from such a bound can be adjusted by setting a parameter $\gamma$, and the bound can also be utilized for learning generative models such as VAEs.

Hierarchical models are widely used in practical Bayesian inference. This work further derives two objectives with fractional posteriors for hierarchical models, an objective with structure and an objective that allows subsampling. There are also cases where components in the objectives can be derived analytically and the work shows three of them.

In the experiments, it is first shown that with proper setting of $\gamma$, the fractional posteriors can achieve the correct coverage given a confidence level, while having an interval similar to that from ELBO. For MNIST image modeling problems, tuning $\gamma$ is also useful for having a better model learning capability. On FashionMNIST, it is shown that the generation quality increases when $\gamma$ is reduced, while retaining the theoretical aspects of a variational lower bound.

## update after rebuttal

I was leaning towards acceptance for this work. I still think this work is worthy of acceptance after reading through other reviewers' comments. I keep my score, but do not increase it because I also foresee huge efforts in the revision.

**Claims And Evidence:**

I think the theoretical claims and empirical evidences are strong.

**Essential References Not Discussed:**

There is another line of works in generalized Bayesian inference with empirical losses and regularizers [1-3]. This work has similarly structured objectives as those works (though they do not produce tight lower bounds). I don't think they should be compared, but they are worth mentioning.

[1] Masegosa, A. (2020). Learning under model misspecification: Applications to variational and ensemble methods. Advances in Neural Information Processing Systems, 33, 5479-5491.

[2] Morningstar, W. R., Alemi, A., & Dillon, J. V. (2022, May). PACm-Bayes: Narrowing the empirical risk gap in the misspecified Bayesian regime. In International Conference on Artificial Intelligence and Statistics (pp. 8270-8298). PMLR.

[3] Lai, J., & Yao, Y. (2024). Predictive variational inference: Learn the predictively optimal posterior distribution. arXiv preprint arXiv:2410.14843.

**Experimental Designs Or Analyses:**

Most experiments are on the effects of changing $\gamma$ in the fractional posterior framework. The designs and analyses are sound to me.

**Methods And Evaluation Criteria:**

- The discussion of the methods is thorough and strong. In addition to the variational objectives, this work also provides recipes for stochastic optimization, as well as a second-step variational objective when the first-step approximate posterior is not normalized.
- For the methodology, only the derivation of $\mathcal{L}^{bh}$ is not clear to me. As in A.3, there are two possible objectives, but the main text uses the one with both $u$ and $u'$ in the integrand. I do not see why this one is chosen.
- For a theoretical paper like this, I think the benchmark datasets are enough to support the claims. However, it would be much better if the comparison with $\beta$-VAE is also demonstrated in the experiments.

**Other Comments Or Suggestions:**

No.

**Other Strengths And Weaknesses:**

No.

**Questions For Authors:**

- Why is the objective with both $u$ and $u'$ chosen for $\mathcal{L}^{bh}$? Is there a practical obstacle that keeps the other from being implemented?

**Relation To Broader Scientific Literature:**

From what I can see, this work extends the generalized variational bounds from Li & Turner, 2016 in the context of generalized Bayesian inference, and is heavily influenced by Yin & Zhou, 2018, for implicit approximate posterior.

**Theoretical Claims:**

- It is neat to have a generalized Bayesian inference objective that is a lower bound to the log marginal. I think this contribution is strong enough to establish the work.
- Section 3.1 contains three cases where components in the objective could be derived analytically. Given that modern Bayesian models can be written as probabilistic programs with program tracing and autodiffs, I suppose most of the derivations could be automated in practice.

---

> ### Author Rebuttal · Authors · 2025-03-28
>
> ### Methods And Evaluation Criteria
> > For the methodology, only the derivation of $\mathcal{L}^{bh}$  is not clear to me. As in A.3, there are two possible objectives, but the main text uses the one with both $\boldsymbol{u}$ and $\boldsymbol{u}$’  in the integrand. I do not see why this one is chosen.
>
> This one is chosen because we have its experimental results at time of submission. We now have experimental results for the alternate and simpler bound in A.3 — the empirical results are very similar (for example, the test objective for $\gamma=0.5$ is $1608.0\pm25.4$), and the implementation is simpler and does not involve triple integrals. We also have proof showing non-degeneracy of the implicit distributions is not neccessary, similar to A.4.2.
>
> We will replace the currently chosen bound with the simpler bound in the main text. The current bound will be placed in the appendix. The essential arguments in the experimental section remain the same.
>
> > For a theoretical paper like this, I think the benchmark datasets are enough to support the claims. However, it would be much better if the comparison with $\beta$-VAE is also demonstrated in the experiments.
>
> The $\beta$-VAE is demonstrated in C.4 for generating images from the prior, and this is referenced from the main text in section 6 (line 409, right column). In addition to the parameters for $\beta$-VAE currently in C.4, we have additionally tried with $\beta$ taking values 5 and $10^2$, giving FIDs 78.4 and 99.1. The conclusions are the same. We will include these additional results in the final version.
>
> Subject to space constraints, we will move some of these results to the main text in the camera-ready version.
>
> Since $\beta$-VAE for fractional posteriors is provably less tight than ELBO, we do not include the results in section 5.2/Table 2.
>
> ### Theoretical Claims
> > Section 3.1 contains three cases where components in the objective could be derived analytically. Given that modern Bayesian models can be written as probabilistic programs with program tracing and autodiffs, I suppose most of the derivations could be automated in practice.
>
> Yes indeed, for the most part. In addition, we believe knowing the derivations and derived formulae have pedagogical value, and can give rise to new algorithms and/or more efficient updates in future work. For example, autodiff will not be able to choose $1/\gamma = 1 + 1/n$ (section 3.1.2); nor is it able to decide to apply $\mathcal{L}_\gamma$ first on $q(\boldsymbol{u})$ and then ELBO for $p(\boldsymbol{c})$ (section 3.1.3), and in this order. After these are determined, and perhaps with further simplification of the bounds and equations, then autodiffs can be applied readily.
>
> ### Essential References Not Discussed:
> We will mention [1]-[3] and relate them to our work.
>
> ### Questions For Authors:
> Answered above.

---

> > ### Comment · Reviewer_H3En · 2025-04-02
> >
> > Thank you for the clarifications and the additional experimental results. I will keep my score.

---

### Decision · Program_Chairs · 2025-05-01

**Decision:**

Accept (poster)

**Comment:**

We thank the authors for their compelling contribution.  Reviewers agreed that this work contains both interesting theoretical claims and compelling empirical evidence.  The main idea is novel, and the authors showcase a variety of settings where it can be applied.  The productive back and forth with reviewer F6u2 clarified a number of key questions, and the authors are encouraged to incorporate these points of clarification into the manuscript.